

**Aqueous Reactions of Organic Triplet Excited States with Atmospheric**
**Alkenes**
Richie Kaur [a,b], Brandi M. Hudson [c], Joseph Draper [a, #], Dean J. Tantillo [c], and Cort Anastasio [*a,b]
[a] Department of Land, Air, and Water Resources, University of California, Davis, California
95616, United States
[b] Agricultural & Environmental Chemistry Graduate Group, University of California, Davis
[c] Department of Chemistry, University of California, Davis, California 95616, United States
[#] Now at the Fresno Metropolitan Flood Control District, Fresno, California 93727, United
States
Correspondence to: C. Anastasio (canastasio@ucdavis.edu)

## 12  Abstract

Triplet excited states of organic matter, a.k.a. "triplets", are formed when brown carbon absorbs
light. While triplets can be important photooxidants in atmospheric drops and particles (e.g., they
rapidly oxidize phenols), very little is known about their reactivity toward many classes of
organic compounds in the atmosphere. Here we measure the bimolecular rate constants of the
triplet excited state of benzophenone ($^3$BP*), a model species, with 17 water-soluble $C_3 - C_6$
alkenes that have either been found in the atmosphere or are reasonable surrogates for identified
species. Measured rate constants ($k_{ALK+3BP*}$) vary by a factor of 30 and are in the range of $(0.24 - $
$7.5) \times 10^9 \ M^{-1} \ s^{-1}$. Biogenic alkenes found in the atmosphere – e.g., cis-3-hexen-1-ol, cis-3-
hexenyl acetate, and methyl jasmonate – react rapidly, with rate constants above $1 \times 10^9 \ M^{-1} \ s^{-1}$.
Rate constants depend on alkene characteristics such as the location of the double bond,
stereochemistry, and alkyl substitution on the double bond. There is a reasonable correlation
between $k_{ALK+3BP*}$ and the calculated one-electron oxidation potential (OP) of the alkenes ($R^2 = $
0.58); in contrast, rate constants are not correlated with bond dissociation enthalpies, bond
dissociation free energies, or computed energy barriers for hydrogen abstraction. Using the OP
relationship, we estimate aqueous rate constants for a number of unsaturated isoprene and



limonene oxidation products with $^3$BP*: values are in the range of (0.080–1.7) $\times$ 10$^9$ M$^{-1}$ s$^{-1}$,
with generally faster values for limonene products. Using our predicted rate constants, along
with values for other reactions from the literature, we conclude that triplets are probably minor
oxidants for isoprene and limonene-related compounds in cloudy or foggy atmospheres, except
in cases where the triplets are very reactive.

**1    Introduction**

Photochemical processes in atmospheric aqueous phases (e.g., cloud and fog drops and

aqueous particles) are important sources and sinks of secondary organic species (Blando and
Turpin, 2000; Lim et al., 2010; Kroll and Seinfeld, 2008; Volkamer et al., 2009; Gelencsér and
Varga, 2005), which represent a large fraction of aerosol mass (Zhang et al., 2007; Hallquist et
al., 2009). Many of these reactions involve photooxidants, including hydroxyl radical ($^\bullet$OH),
which is widely considered to be the dominant aqueous oxidant (Herrmann et al., 2010;
Herrmann et al., 2015). But there are numerous other aqueous photooxidants, such as singlet
molecular oxygen, hydroperoxyl radical/superoxide radical anion, hydrogen peroxide, and triplet
excited states of organic matter ($^3$C* or triplets) (Lee et al., 2011; Anastasio and McGregor,
2001; Kaur and Anastasio, 2017; Anastasio et al., 1996; Anastasio et al., 1994; Zepp et al., 1977;
Wilkinson et al., 1995; Kaur and Anastasio, 2018). Formed from the photoexcitation of colored
organic matter (i.e., brown carbon), triplets are important oxidants in surface waters for several
classes of organic compounds, including phenols, anilines, amines, phenylurea herbicides, and
heterocyclic sulfur-containing compounds (Canonica et al., 1995; Canonica and Hoigné, 1995;
Arnold, 2014; Canonica et al., 2006a; Bahnmüller et al., 2014; Boreen et al., 2005); however,
very little is known about atmospheric triplets.



Recent studies have shown that aqueous triplets can be the dominant oxidants for phenols
emitted during biomass combustion (Smith et al., 2014), with phenol lifetimes on the order of a
few hours in fog drops (Kaur and Anastasio, 2018) and aqueous particle extracts (Kaur et al.,
2018).There is also evidence that triplets can oxidize some unsaturated aliphatic compounds.
Richards-Henderson et. al.(Richards-Henderson et al., 2014b) measured rate constants for five
unsaturated biogenic volatile organic compounds (BVOCs) with the model triplets 3,4-
dimethoxybenzaldhyde and 3'-methoxyacetophenone, and found that rate constants ranged
between $10^7$ and $10^9$ $M^{-1}\,s^{-1}$. Other laboratory studies have shown that triplet states of
photosensitizers such as imidazole-2-carboxaldehyde and 4-benzoylbenzoic acid can oxidize
gaseous aliphatic BVOCs, e.g., isoprene and limonene, and model aliphatic compounds, e.g., 1-
octanol, at the air-water interface to form low-volatility products that increase particle mass (Fu
et al., 2015; Rossignol et al., 2014; Li et al., 2016; Laskin et al., 2015). However, the
atmospheric importance of these types of processes are unclear (Tsui et al., 2017). Additionally,
we recently reported that natural triplets in illuminated fog waters and particle extracts are
significant oxidants for methyl jasmonate, an unsaturated aliphatic BVOC, accounting for 30–80
% of its aqueous loss during illumination (Kaur et al., 2018; Kaur and Anastasio, 2018).
Abundant BVOCs such as isoprene and limonene are rapidly oxidized in the gas phase to
form unsaturated $C_3$–$C_6$ oxygenated volatile organic compounds (OVOCs) that include isoprene
hydroxyhydroperoxides, isoprene hydroxynitrates, and isoprene and limonene aldehydes (Surratt
et al., 2006; Paulot et al., 2009b; Crounse et al., 2011; Ng et al., 2008; Walser et al., 2008; Paulot
et al., 2009a). Several of these first-generation products have high Henry's law constants, above
$10^4$ $M\,atm^{-1}$ (Marais et al., 2016) and partition significantly into cloud and fog drops and, to a
smaller extent, into aerosol liquid water. There, they can undergo further oxidation by aqueous
photooxidants, including $^\bullet OH$ and ozone (Wolfe et al., 2012; St. Clair et al., 2015; Khamaganov
and Hites, 2001; Schöne and Herrmann, 2014; Lee et al., 2014) and possibly triplets. Our past



measurements have shown that steady-state concentrations of $^3C^*$ are orders of magnitude higher
than $^\bullet OH$ in fog waters and aqueous particles (Kaur et al., 2018; Kaur and Anastasio, 2018) and
thus they might contribute significantly to the loss of OVOCs derived from isoprene and other
precursors. However, testing this hypothesis requires rate constants for the reactions of triplets
with alkenes, which are scarce.

To address this gap, we studied the reactions of 17 $C_3 - C_6$ unsaturated compounds with

the triplet state of the model compound benzophenone (Fig. 1). While our 17 unsaturated
compounds include alcohols, esters, and chlorinated compounds, for simplicity we refer to them
all as "alkenes". The tested alkenes include BVOCs emitted into the atmosphere as well as
surrogates for some of the small unsaturated gas-phase products formed as secondary OVOCs.
The goals of this study are to: 1) measure rate constants for reactions of the alkenes with the
triplet excited state of benzophenone, 2) explore quantitative structure-activity relationships
(QSARs) between the measured rate constants and calculated alkene properties (e.g., the one-
electron oxidation potential) and 3) use a suitable QSAR to estimate rate constants for triplets
with some unsaturated isoprene and limonene oxidation products to predict whether or not
triplets are significant oxidants for these species in cloud and fog drops.

**2    Methods**
**2.1 Chemicals**

All chemicals were purchased from Sigma-Aldrich with purities of 95 % and above, and

were used as received: the compound numbers, compound names, and abbreviated names are
listed in Table 1. All chemical solutions were prepared using purified water (Milli-Q water) from
a Milli-Q Plus system (Millipore; ≥18.2 MΩ cm) with an upstream Barnstead activated carbon



cartridge. The pH of each reaction solution was adjusted to 5.5 (± 0.2) using a 1.0 mM phosphate
buffer.

## 2.2 Kinetic Experiments

Bimolecular rate constants of the alkenes with the triplet state of benzophenone ($^3$BP*)

were measured using a relative rate technique, as described in in the literature (Richards-
Henderson et al., 2014a; Finlayson-Pitts and Pitts Jr, 1999). The technique involves illuminating
a solution containing the triplet precursor (BP), a reference compound with a known second-
order rate constant with $^3$BP*, and one test alkene for which the rate constant is unknown.
Buffered, air-saturated solutions containing 50 µM each of the reference and test compounds and
100 µM of BP were prepared and then 10 mL of this solution was illuminated in a stirred 2-cm,
air-tight quartz cuvette (Spectrocell) at 25 °C. Samples were illuminated with a 1000 W Xenon
arc lamp filtered with an AM 1.0 air mass filter (AM1D-3L, Sciencetech) and 295 nm long-pass
filter (20CGA-295, Thorlabs) to mimic tropospheric solar light (Fig. S1 of the Supplemental
Information). At various intervals, aliquots of illuminated sample were removed and analyzed for
the concentration of reference compound and test alkene using HPLC (Shimadzu LC-10AT
pump, ThermoScientific BetaBasic-18 C$_{18}$ column (250 × 33 mm, 5 µM bead), and Shimadzu-
10AT UV-Vis detector). Parallel dark controls were employed with every experiment using an
aluminum foil-wrapped cuvette containing the same solution in the illumination chamber and
analyzed in the same manner as the illuminated solutions.

In every case, loss of test and reference compounds followed first-order kinetics. Plotting

the change in concentration of the test alkene against that of the reference compound yields a
linear plot that is represented by:
$$\ln\frac{[\text{Reference}]_0}{[\text{Reference}]_t} = \frac{k_{\text{Reference}+3\text{BP*}}}{k_{\text{ALK}+3\text{BP*}}}\ \ln\frac{[\text{ALK}]_0}{[\text{ALK}]_t} \qquad\qquad (1)$$



where [Reference]$_0$, [Reference]$_t$, [ALK]$_0$, and [ALK]$_t$ are the concentrations of the reference
and test alkenes at times zero and $t$, respectively, and  $k_{\text{Reference+3BP*}}$ and $k_{\text{ALK+3BP*}}$ are the
bimolecular rate constants for the reaction of the reference and test alkenes with $^3$BP*,
respectively. A plot of Eq. (1) (with the y-intercept fixed at the origin) gives a slope equal to the
ratio of the bimolecular rate constants; dividing $k_{\text{Reference+3BP*}}$ by the slope gives $k_{\text{ALK+3BP*}}$. The
measurement technique is illustrated in Fig. S2.

**2.3 Calculation of Oxidation Predictor Variables**

All calculations were completed using the Gaussian 09 software suite (Frisch et al.,
2009). Structures of alkenes were optimized using uB3LYP/6-31+G(d,p) (Becke, 1992, 1993;
Lee et al., 1988; Stephens et al., 1994; Tirado-Rives and Jorgensen, 2008) for reaction coordinate
calculations and the CBS-QB3 (Montgomery Jr et al., 1999) method for predicting bond
dissociation enthalpies (BDEs), bond dissociation free energies (BDFEs), and oxidation
potentials (OPs). Transition state structures (TSSs) were optimized in the triplet state using
uB3LYP/6-31+G(d,p) (Becke, 1992, 1993; Lee et al., 1988; Stephens et al., 1994; Tirado-Rives
and Jorgensen, 2008). TSSs were confirmed by the presence of an imaginary frequency and
minima (reactants and products) were confirmed by the absence of imaginary frequencies. Free
energy ($\Delta G$) and enthalpy ($\Delta H$) differences were determined by comparing TSS energies relative
to their reactant energies. For solvent phase calculations, the solvent model density (SMD)
(Marenich et al., 2009) continuum model was used with water as the solvent. To calculate BDEs,
the neutral (AH) and radical species (A$^\bullet$) (resulting from H atom abstraction) of each alkene and
the H radical (H$^\bullet$) were optimized in the gas phase. Using the computed enthalpies (H) and Eq.
(2), the predicted BDEs of each alkene were determined.
$BDE = H_{A^{\cdot}} + H_{H^{\cdot}} - H_{AH}$ (2)





To determine the predicted BDFEs, the neutral ($AH_g$, $AH_{aq}$) and radical species ($A^\bullet_g$,
$A^\bullet_{aq}$) of each alkene and the H radical ($H^\bullet_g$, $H^\bullet_{aq}$) were optimized in the gas and solvent phases
and their differences taken to give $\Delta G^\circ_{solv,AH}$, $\Delta G^\circ_{solv,A^\bullet}$, and $\Delta G^\circ_{solv,H^\bullet}$, respectively. Based on
the thermodynamic cycle shown (Scheme 1), these values were used in Eqs. (3) and (4) to
calculate the BDFEs of C–H and O–H bonds.

$$AH_g \xrightarrow{\Delta G_{rxn,g}} A^\bullet_g \; + \; H^\bullet_g$$
$$\Big\downarrow \Delta G^\circ_{solv,A} \qquad\qquad \Big\downarrow \Delta G^\circ_{solv} \quad \Big\downarrow \Delta G^\circ_{solv}$$
$$AH_{aq} \xrightarrow{\Delta G_{rxn,aq}} A^\bullet_{aq} \; + \; H^\bullet_{aq}$$


**Scheme 1.** Thermodynamic cycle used to calculate bond dissociation free energies.
$\Delta\Delta Gsolv = \Delta G^\circ_{solv,A^\cdot} + \Delta G^\circ_{solv,H^\cdot} - \Delta G^\circ_{solv,AH}$                 (3)
$\Delta G^\circ_{ox} = \Delta G_g + \Delta\Delta G_{solv}$                 (4)
To predict OPs, the neutral ($A_g$, $A_{aq}$) and radical cation ($A^{\bullet+}_g$, $A^{\bullet+}_{aq}$) forms of each alkene
were optimized in the gas and solvent phase, their difference giving $\Delta G^\circ_{solv,A}$ and $\Delta G^\circ_{solv,A^\bullet}$.
Based on the thermodynamic cycle shown below (Scheme 2), these values were used in Eqs. (5–
7) to calculate the OP of each alkene.

$$A_g \xrightarrow{IE_{gas}} A^{\bullet+}_g \; + \; e^-_g$$
$$\Big\downarrow \Delta G^\circ_{solv,A} \qquad \Big\downarrow \Delta G^\circ_{solv,A^{\bullet+}} \Big\downarrow \Delta G=0$$
$$A_{aq} \xrightarrow{\Delta G^\circ_{ox,aq}} A^{\bullet+}_{aq} \; + \; e^-_g$$


**Scheme 2.** Thermodynamic cycle used to calculate oxidation potentials.
$\Delta\Delta G_{solv} = \Delta G^\circ_{solve,A^\cdot+} - \Delta G^\circ_{solv,A}$                 (5)
$\Delta G^\circ_{ox} = IE_{gas} + \Delta\Delta G_{solv}$                 (6)





$$E_{ox} = -\left(\frac{-\Delta G^{\circ}_{ox}}{nF} + SHE\right) \qquad (7)$$
where $n$ is the number of electrons, $F$ is Faraday's constant (96485.3365 C/mol), and $SHE$ is the
potential of the standard hydrogen electrode (4.28 V).

Subsequent MP2/CBSB3 (Petersson et al., 1988; Petersson and Al-Laham, 1991;

Petersson et al., 1991; Mayer et al., 1998) single point calculations were used to compute the
highest occupied molecular orbitals (HOMOs) and singly occupied molecular orbitals (SOMOs).
Structural drawings were produced using CYLView (Legault, 2009).

## 3    Results and discussion

### 3.1 Alkene-triplet bimolecular rate constants ($k_{ALK+3BP*}$)

Fig. 1 shows the chemical structures for all 17 alkenes and the triplet precursor,

benzophenone. The alkenes have molecular weights ranging between 58 and 220 g mol$^{-1}$ and
include 13 alcohols, three esters and one chlorinated compound. The model triplet precursor
benzophenone (BP) has been previously employed in surface water studies, and rapidly reacts
with aromatics such as substituted phenols and phenyl urea herbicides with rate constants faster
than $10^9$ M$^{-1}$ s$^{-1}$ (Canonica et al., 2000; Canonica et al., 2006b).

The bimolecular rate constants for the alkenes with the excited triplet state of BP

($k_{ALK+3BP*}$) vary by a factor of 30, spanning the range of $(0.24$–$7.5) \times 10^9$ M$^{-1}$s$^{-1}$. Values are
shown in Tables 1 and S1, and in Fig. S3, where the alkenes are numbered in ascending order of
their reactivity towards $^3$BP*. Based on their rate constants, the alkenes appear to be broadly split
into two groups – the slower alkenes (**1–9**), whose rate constants lie below $1 \times 10^9$ M$^{-1}$s$^{-1}$ and
span a range of only a factor of 2.5, and the faster alkenes (**10–17**) which vary by a factor of 5.
Notably, three of the four BVOCs identified in emissions to the atmosphere – 3MBO (**12**), cHxO



(**15**), cHxAc (**16**) and MeJA (**17**) – react rapidly with $^3$BP\*, with rate constants greater than $1 \times$
$10^9 \ \mathrm{M^{-1} \ s^{-1}}$.

Three alkene characteristics appear to increase reactivity: internal (rather than terminal)

double bonds; methyl substitution on the double bond; and alkene stereochemistry. To more
specifically examine the impact of these variables, we compare the rate constants for three sets of
alkenes (Fig. 2). The lowest free energy and enthalpy barriers for the abstraction of a hydrogen
atom are also shown in Fig. 2 (and in Table 1); while overall these computed barriers are not
well-correlated with rate constants (discussed below), lower barriers generally correspond to
higher rate constants for the sets of alkenes in Fig. 2. Based on the first two sets of compounds in
Fig. 2, internal alkenes react faster with $^3$BP\* than do terminal isomers. For example, cHxAc
(**16**), an internal hexenyl acetate, has a reaction rate constant 11 times faster than its terminal
isomer 5HxAc (**9**). The corresponding alcohols also exhibit the same trend: the internal alkenes
cHxO (**15**) and tHxO (**10**) react 27 and 5.8 times faster, respectively, than the terminal isomer
5HxO (**1**). This dependence of reactivity on double bond location has implications for isoprene
hydroxyhydroperoxides (ISOPOOH) and hydroxynitrates (ISONO$_2$), which have both terminal
(β-) and non-terminal (δ-) isomers formed from gas-phase oxidation (Marais et al., 2016; Paulot
et al., 2009b; Paulot et al., 2009a). Based on our results we expect that the δ-isomers react more
quickly with organic triplets than the β-isomers.

Alkene stereochemistry also affects the triplet-alkene reaction rate constant. The data in

the middle of Fig. 2 shows that cis-HxO (**15**) reacts nearly five times more quickly with $^3$BP\*
than does trans-HxO (**10**), consistent with the lower predicted energy barrier for hydrogen atom
abstraction from the cis isomer. Finally, addition of electron-donating substituents (methyl
groups) on an unsaturated carbon atom also increases the rate constant. This is evident from
comparing 2B1O (**8**) and its methyl-substituted analog 3MBO (**12**): $k_{\mathrm{ALK+3BP*}}$ is 3.7 times faster
with the methyl group (Fig. 2). Mechanistically, triplet-induced oxidation can proceed via either
hydrogen atom transfer or a proton-coupled-electron transfer (Canonica et al., 1995; Warren et



al., 2010; Erickson et al., 2015) and the presence of an electron-donating substituent on the
double bond likely selectively stabilizes the intermediates (e.g., radical or radical cation) formed
from these two processes, as well as the transition state structures for their formation.

**3.2 Relationship between *k* and one-electron oxidation potential**

Our next goal was to develop a quantitative structure-activity relationship (QSAR) so that

we can predict rate constants for alkene-triplet reactions. To use as predictor variables in the
QSARs we computed several properties of the alkenes: bond dissociation enthalpy and free
energy for various hydrogen atoms (Fig. S4), free energy and enthalpy barriers for hydrogen
atom abstraction (Table 1), and one-electron oxidation potentials (Table 1). Apart from the
oxidation potential, none of the other properties correlate well with the measured rate constants
(Figs. S5 and S6). While there is no correlation between the rate constants and predicted energy
barriers, alkenes with lower predicted free energy barriers ($\Delta G^{\ddagger}$) are predicted to be fast-reacting,
with rate constants above $5 \times 10^8$ M$^{-1}$ s$^{-1}$ (Fig. S6). As shown in Fig. S6, computed barriers
predict a much larger variation in rate than observed experimentally, suggesting that the breaking
of the C–H or O–H bond does not occur in the rate-determining step for all alkenes.

Of all the properties examined, the one-electron oxidation potential (OP) of the alkenes

best correlates with the (log of) measured rate constants, with rate constants generally increasing
as the alkenes are more easily oxidized, i.e., at lower OP values ($R^2 = 0.58$) (Fig. 3). Measured
rate constants for 13 of the 16 alkenes lie within (or very near to) the 95 % confidence interval
(blue lines) of the regression fit, but there are three notable outliers: hexen-1,3-diol (**3**, HDO),
cis-3-hexen-1-ol (**15**, cHxO) and cis-3-hexenylacetate (**16**, cHxAc). The measured HDO rate
constant is 3.3 times lower than that predicted by the regression line, while measured rate
constants for cHxO and cHxAc are 3.9 and 4.9 times higher, respectively, than predicted.

To try to assess why these compounds differ from the others, we calculated the highest

occupied molecular orbital (HOMO) of the alkene and the singly occupied molecular orbital
(SOMO) of the alkene radical cation (i.e., after oxidation) (Fig. 4). Depending on the system,



oxidation is predicted to occur by removing an electron either from the π system of the C–C
double bond or from a lone pair on the O atom. This is illustrated in Fig. 4, which shows the
HOMO and SOMO structures for HDO (**3**), where the electron is removed from the C-C double
bond, and 3B1O (**5**), where the electron is removed from the oxygen atom. However, the three
outliers in the correlation do not all fall into just one of these categories: for cHxAc (**16**) the
electron is most likely abstracted from the oxygen, while for HDO (**3**) and cHxO (**15**) the
electron is likely removed from the π system (Tables S2 and S3). This suggests that the location
of electron removal does not control the rate constants. We also examined if the rate of loss of
cHxO might be enhanced due to oligomerization, where an initially formed cHxO radical leads
to additional cHxO loss. Since the pseudo-first-order rate constant of oligomerization should
increase with initial cHxO concentration, we measured the rate constant for cHxO loss over a
range of initial concentrations (2–50 μM). However, as shown in Fig. S8, the rate constant for
cHxO loss does not depend on its concentration, suggesting that oligomerization is an
unimportant loss process for cHxO in our experiments. Thus, it is not clear why these three
compounds do not fall closer to the regression line of Fig. 3. However, except for **16**, all of the
alkenes fall within a factor of four of the correlation line (grey lines). Finally, even though there
is a good correlation between rate constant and OP in Fig. 3, it does not indicate whether these
reactions proceed via pure electron transfer, proton-coupled electron transfer, or hydrogen
transfer. As discussed earlier, since the predicted energy barriers for hydrogen abstraction do not
correlate with measured rate constants (Fig. S6) and appear to split into two groups, there
remains uncertainty about the mechanism of triplet-induced oxidation of the alkenes.

**3.3 Predicted triplet-OVOC bimolecular rate constants**

We next use the relationship in Fig. 3, along with calculated oxidation potentials, to
predict second-order rate constants for $^3$BP* with a set of unsaturated isoprene- and limonene-
derived OVOCs. As shown in Fig. 5, we predict that limonene products generally react faster



with $^3$BP* than do isoprene products. For the five isoprene-derived OVOCs that we considered,
rate constants vary by a factor of 17, and range between $(0.080–1.4) \times 10^9$ M$^{-1}$ s$^{-1}$ (Table 1, Fig.
5). The δ-isomers of ISOPOOH and ISONO$_2$, which contain internal double bonds, have lower
computed one-electron oxidation potentials and thus higher predicted rate constants compared to
the terminal β-isomers. This is similar to the trend observed with the other alkenes (Fig. 2). In
case of isoprene hydroperoxyaldehydes, we were able to determine the oxidation potential for
only HPALD2 (**22**), and its predicted reaction rate constant ($\pm$ 1 SE) of $4.0$ ($\pm$ $0.9$) $\times 10^8$ M$^{-1}$ s$^{-1}$
is among the lowest of the isoprene-derived alkenes (Fig. 5).

We calculated OP values and triplet rate constants for three limonene-derived OVOCs:

limonene aldehyde (LMNALD) and two dihydroxy-limonene aldehydes (2,5OH-LMNALD and
4,7OH-LMNALD). Compared to the isoprene-derived alkenes, the rate constants for all three
limonene products are high, and range between $(0.89–1.7) \times 10^9$ M$^{-1}$ s$^{-1}$. All of the limonene
aldehydes (as well as the isoprene products) can have several isomers whose calculated oxidation
potentials can vary, which affects the predicted rate constant. For example, for 4,7OH-LMNALD
(**25**) the computed oxidation potential for five of its isomers vary between 2.17 and 2.48 V
(Table S4), which leads to a relative standard deviation of 40 % in the predicted rate constants
for the various isomers. For each OVOC, the predicted rate constants in Table 1 are for the
lowest energy isomers whose structures are shown in Fig. S9.
**3.4 Role of triplets in the fate of isoprene- and limonene-derived OVOCs**

Next, we use our estimated rate constants, along with previously published estimated

values for rates of other loss processes (Table S5), to understand the importance of triplets as
sinks for isoprene- and limonene-derived OVOCs in a foggy/cloudy atmosphere. For our simple
calculations we use a liquid water content of $1 \times 10^{-6}$ L-aq/L-g, a temperature of 25 °C, and
calculated Henry's law constants from EPISuite (US EPA. Estimation Programs Interface
Suite™ for Microsoft® Windows v 4.1, 2016) (Table S6).  From these inputs, we estimate that



between 10 and 97 % of the OVOCs will be partitioned into the aqueous phase under our
conditions (Table S6). The OVOC sinks we consider are photolysis and reactions with hydroxyl
radical ($^{\bullet}$OH) and ozone ($O_3$) in the gas phase as well as hydrolysis and reactions with $^{\bullet}$OH, $O_3$,
and triplets in the aqueous phase (Table S5). Based on typical oxidant concentrations in both
phases and available rate constants with sinks, the overall pseudo-first-order rate constants for
OVOC loss are estimated to be in the range of $(0.27–3.0) \times 10^{-4}\,\mathrm{s}^{-1}$, corresponding to overall
lifetimes of 0.93 to 10 h (Table S7). The only exception is δ-ISONO2, which is expected to
undergo rapid hydrolysis to form its corresponding diol (Jacobs et al., 2014) with a lifetime of
just 0.078 h (280 s).

Fig. 6 shows the overall loss rate constants, and the contribution from each pathway, for

four of these OVOCs: δ4-ISOPOOH (**19**), β-ISONO2 (**20**), HPALD2 (**22**) AND 4,7-OH
LMNALD (**25**). Overall, aqueous-phase processes dominate the fate of these OVOCs,
accounting for the bulk of their loss; but the contribution of aqueous triplets to OVOC loss
depends strongly on the triplet reactivity. Panel (a) of Fig. 6 shows OVOC loss when we assume
that the aqueous triplets are highly reactive, i.e., using rate constants estimated for $^3$BP* (Fig. 5).
Since our recent measurements (Kaur et al., 2018; Kaur and Anastasio, 2018) indicate that, on
average, ambient triplets are not this reactive, this scenario likely represents an upper bound for
the triplet contribution. In this case triplets are the dominant sinks for δ4-ISOPOOH and 4,7-OH
LMNALD, accounting for 74 % and 47 % of their total losses, respectively (Fig. 6a). For β-
ISONO$_2$ and HPALD2, triplets are not dominant but still significant, accounting for 19 % and 24
% of loss, respectively, while other sinks dominate. For the OVOCs where we calculated rate
constants with $^3$BP* (Fig. 5) but that are not shown in Fig. 6, the triplet contribution varies
widely, from less than 1 % for δ-ISONO2 (**21**), where hydrolysis dominates, to 59 % for 2,5-OH
LMNALD (**24**) (Table S7).





While $^3$BP* likely represents an upper-bound of triplet reactivity in atmospheric waters,
our recent measurements indicate that the triplets in fog waters and particles have an average
reactivity that is typically similar to 3'-methoxyacetophenone (3MAP) and 3,4-
dimethoxybenzaldehyde (DMB) (Kaur et al., 2018; Kaur and Anastasio, 2018). A comparison of
our $^3$BP* rate constants (Table 1) with the average values for the 3MAP and DMB triplets for a
subset of the alkenes (Richards-Henderson et al., 2014b) indicates that the  average 3MAP/DMB
triplet rate constants are 1–18 % of the corresponding $^3$BP* values.  Thus to scale alkene-triplet
rate constants from $^3$BP* to the 3MAP and DMB triplets we take the median value of 4 %, which
is derived from the MeJA rate constants (Table S8). Fig. 6b shows the calculated fates of the
OVOCs in the case where we consider "typical reactivity" triplets, i.e., where we multiply the
$^3$BP* + OVOC rate constants (Fig. 5) by a factor of 0.04. Under these conditions, triplets are
minor oxidants (Fig. 6b), accounting for 9 % and 3 % of the loss of δ4-ISOPOOH and 4,7-
LMNALD, respectively, and approximately 1 % for the other two OVOCs. This suggests that
aqueous triplets are generally minor sinks for OVOCs derived from isoprene and limonene, in
contrast to the case for phenols, where triplets appear to be the major sink (Smith et al., 2014; Yu
et al., 2014; Kaur and Anastasio, 2018). However, there are several important uncertainties in
our determination that triplets are likely minor sinks for oxygenated alkenes. First, the factor we
used to adjust from $^3$BP* rate constants to triplet 3MAP/DMB rate constants (i.e., a factor of
0.04) is quite uncertain: values for the three BVOCs examined range from 0.01 to 0.18 (Table
S8). Additionally, there are very few measurements of triplets in atmospheric drops or particles
(Kaur et al., 2018; Kaur and Anastasio, 2018), and only from two sites, so it is possible that we
are underestimating the average reactivity and/or concentrations of triplets in atmospheric drops
and particles.



## 4 Conclusions

To explore whether triplet excited states of organic matter might be important sinks for unsaturated organic compounds, we measured rate constants for 17 $C_3$–$C_6$ alkenes with the triplet excited state of benzophenone ($^3$BP*). The resulting bimolecular rate constants span the range of $(0.24–7.5) \times 10^9$ $M^{-1}s^{-1}$. Notably, the rate constants are high (above $10^9$ $M^{-1}s^{-1}$) for some important green leaf volatiles emitted from plants – 3MBO, cHxO, cHxAc, and MeJA. Rate constants appear to be enhanced by alkene characteristics such as an internal double bond, cis-stereochemistry, and alkyl substitution on the double bond.

To be able to predict rate constants for other alkenes, we examined QSARs between our measured rate constants and a variety of calculated properties for the alkenes and $^3$BP*-alkene transition states. Rate constants are not correlated with bond dissociation enthalpies, free energies or predicted energy barriers for removal of various hydrogen atoms, but are reasonably correlated with the one-electron oxidation potential of the alkenes ($R^2 = 0.58$). Based on the relationship between rate constants and oxidation potential, we predict that highly reactive triplets will react with first generation isoprene- and limonene- oxidation products with rate constants on the order of $10^8$–$10^9$ $M^{-1}$ $s^{-1}$, with higher values for the δ–isomers compared to terminal β–isomer products. Using these rate constants in a simple model of OVOC chemistry in a foggy/cloudy atmosphere suggests that highly reactive aqueous triplets could be significant oxidants for some isoprene hydroxyhydroperoxides and limonene aldehydes. However, for our current best estimate of typical reactivities, triplets are a minor sink for isoprene- and limonene- derived OVOCs.

To more specifically quantify the contributions of triplet excited states towards the loss of alkenes in particles and drops requires more insight into both the reactivities and concentrations of atmospheric triplet species. In addition, assessing whether triplets might be important sinks for



other organic species requires more measurements of reaction rate constants with
atmospherically relevant organics.
**Competing Interests**
The authors declare that they have no conflict of interest.
**Author Contribution**
CA and RK conceptualized the research goals and designed the experiments. RK and JD
performed the laboratory work, while BH and DT planned and performed the computational
calculations. RK analyzed the experimental data and prepared the manuscript with contributions
from all co-authors, particularly BH, who wrote the sections on computational calculations and
prepared the corresponding figures. CA reviewed and edited the manuscript. CA and DT
provided oversight during the entire process.
**Data Availability**
Data are available upon request.
**Acknowledgements**
We thank Jacqueline R. Labins for assistance with rate constant measurements and Tran Nguyen
for helpful discussions on the reactivity of isoprene oxidation products. This research was funded
by the National Science Foundation (Grants AGS-1105049 and AGS-1649212), the University
of California - Davis Office of Graduate Studies, a UC Guru Gobind Singh Fellowship, and the
Agricultural and Environmental Chemistry Graduate Group at UC Davis.



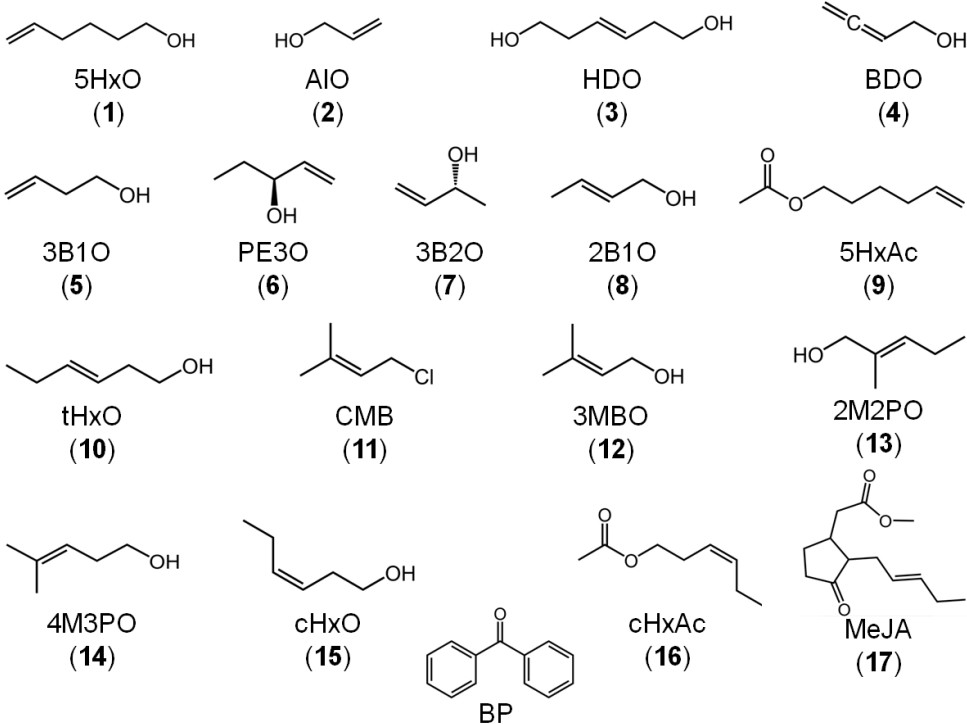

**Fig. 1** Chemical structures of the reactant species used in this study: 17 alkenes and one model triplet, benzophenone. Compound numbers are in parentheses.





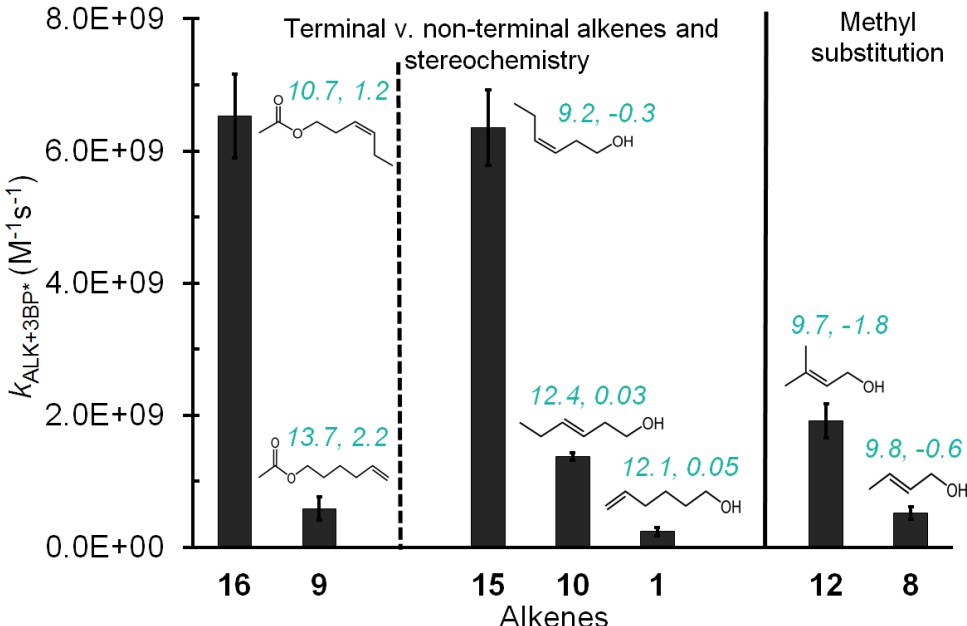

**Fig. 2** Comparison of three sets of alkenes to illustrate how rate constants with the benzophenone
triplet state vary with double bond location, stereochemistry, and methyl substitution. The teal
numbers on each alkene represent the lowest free energy ($\Delta G^{\ddagger}$) and enthalpy ($\Delta H^{\ddagger}$) transition
state barriers in kcal mol$^{-1}$ for H-abstraction by triplet benzophenone; these were calculated at
the uB3LYP/6-31+G(d,p) level of theory. Though computed barriers (Table 1) are not correlated
with the overall rates measured, they broadly match the rate trends within a given set of alkenes
in this figure.





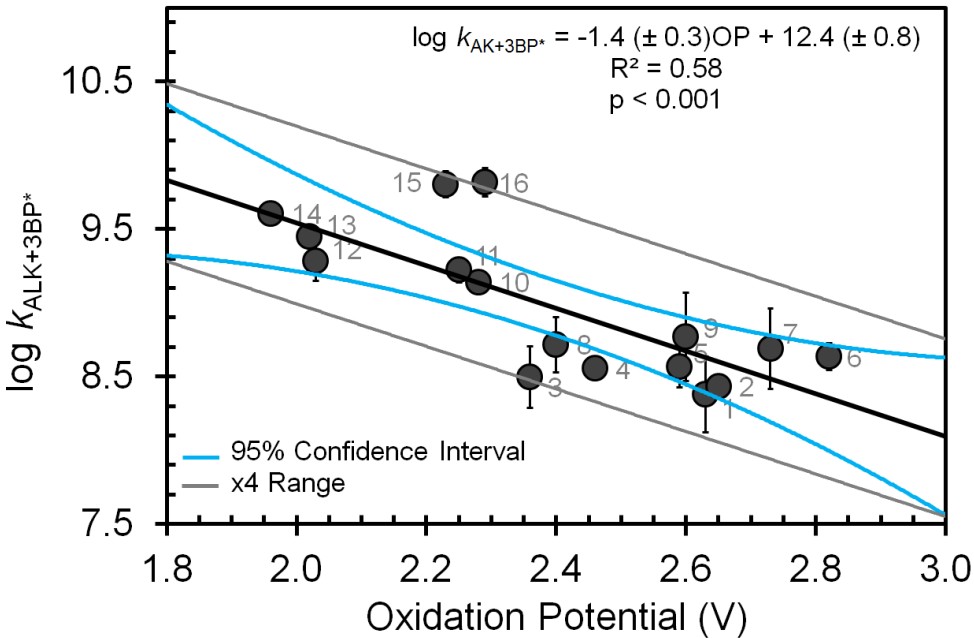

**Fig. 3** Correlation between measured bimolecular rate constants for alkenes with the triplet
excited state of benzophenone ($k_{ALK+3BP*}$) and the computed one-electron oxidation potentials of
the alkenes. Numbers on each point represent the alkene numbers in Table 1. Blue lines represent
95 % confidence intervals of the regression prediction. The gray lines bound the region that is
within a factor of four of the regression prediction; all but one of the alkene values fall within
this. Methyl jasmonate (**17**) is not included in this figure due to computational challenges in
calculating its OP (see Table 1).

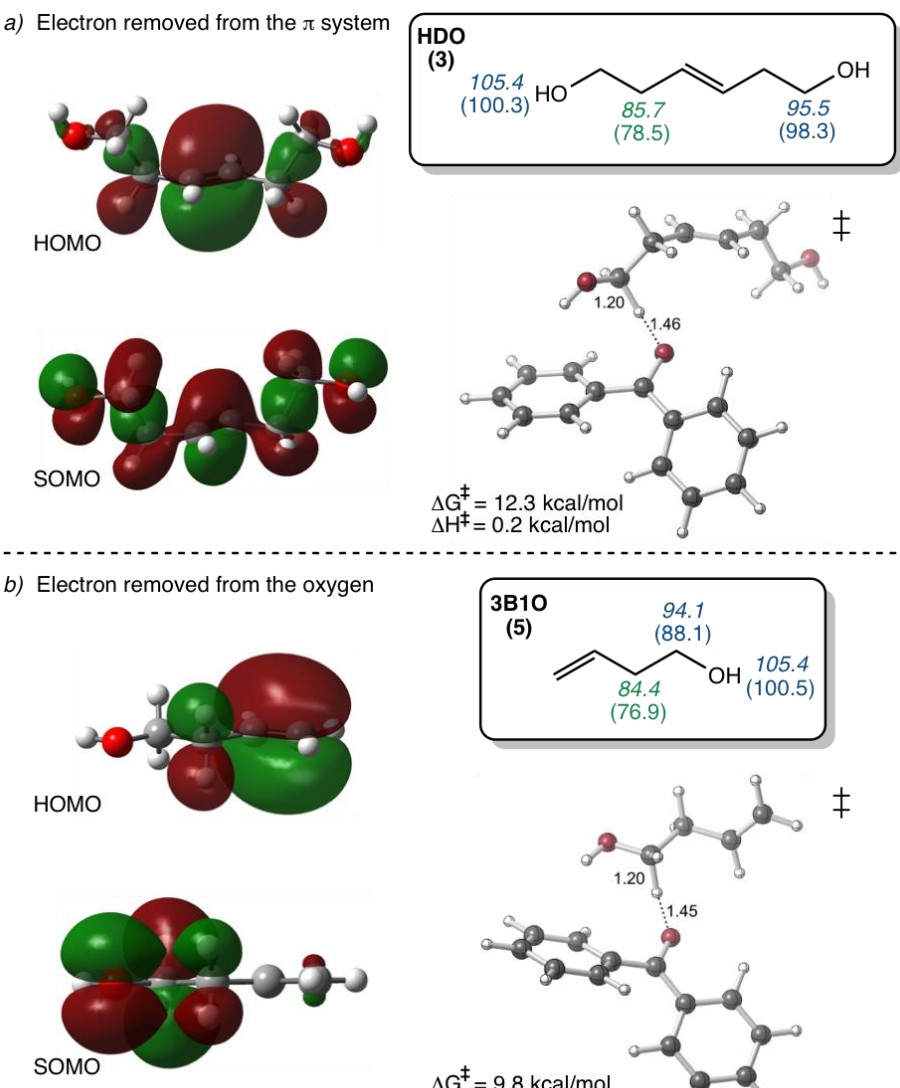

**Fig. 4** Diagrams of the highest occupied molecular orbitals (HOMOs) of the alkenes before
oxidation, and the singly occupied molecular orbitals (SOMOs) after removal of one electron
from the alkenes, and lowest energy transition state structures (‡) of alkenes **3** and **5**. Bond
dissociation enthalpy (italicized) and free energy (in parentheses) for various hydrogen atoms (in
kcal mol$^{-1}$) for each alkene are shown in the boxes. Numbers in green are the lowest values, and
thus represent the most labile hydrogen in each alkene. *a)* The electron removed during H-
abstraction of HDO is predicted to come from the π system, but this results in delocalization due
to hyperconjugation. *b)* The electron removed from 3B1O during H-abstraction is predicted to
come from the oxygen. See SI Tables S2 and S3 for HOMO/SOMO structures and Fig. S4 for
the bond dissociation enthalpies and free energies for other alkenes.



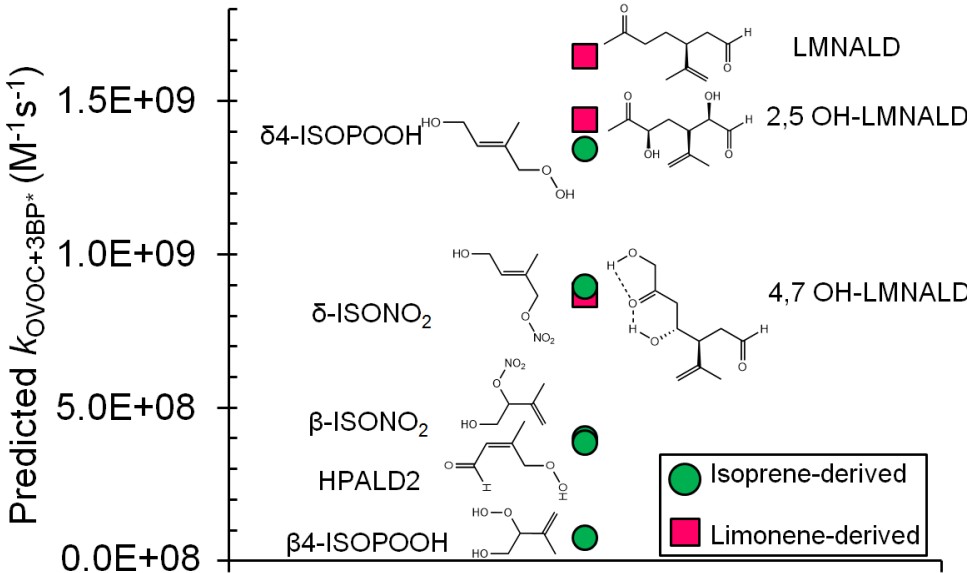



**Fig. 5** Predicted bimolecular rate constants for a range of limonene and isoprene oxidation
products (OVOCs) with the triplet state of BP. Rate constants are estimated from the QSAR with
one-electron oxidation potentials (OPs) (Fig. 3). Oxidation potentials used to predict the rate
constants here (and in Table 1) are for the lowest energy isomers of the OVOCs, which are the
structures shown here. The structures of some of the other, higher energy, isomers are shown in
Table S4.





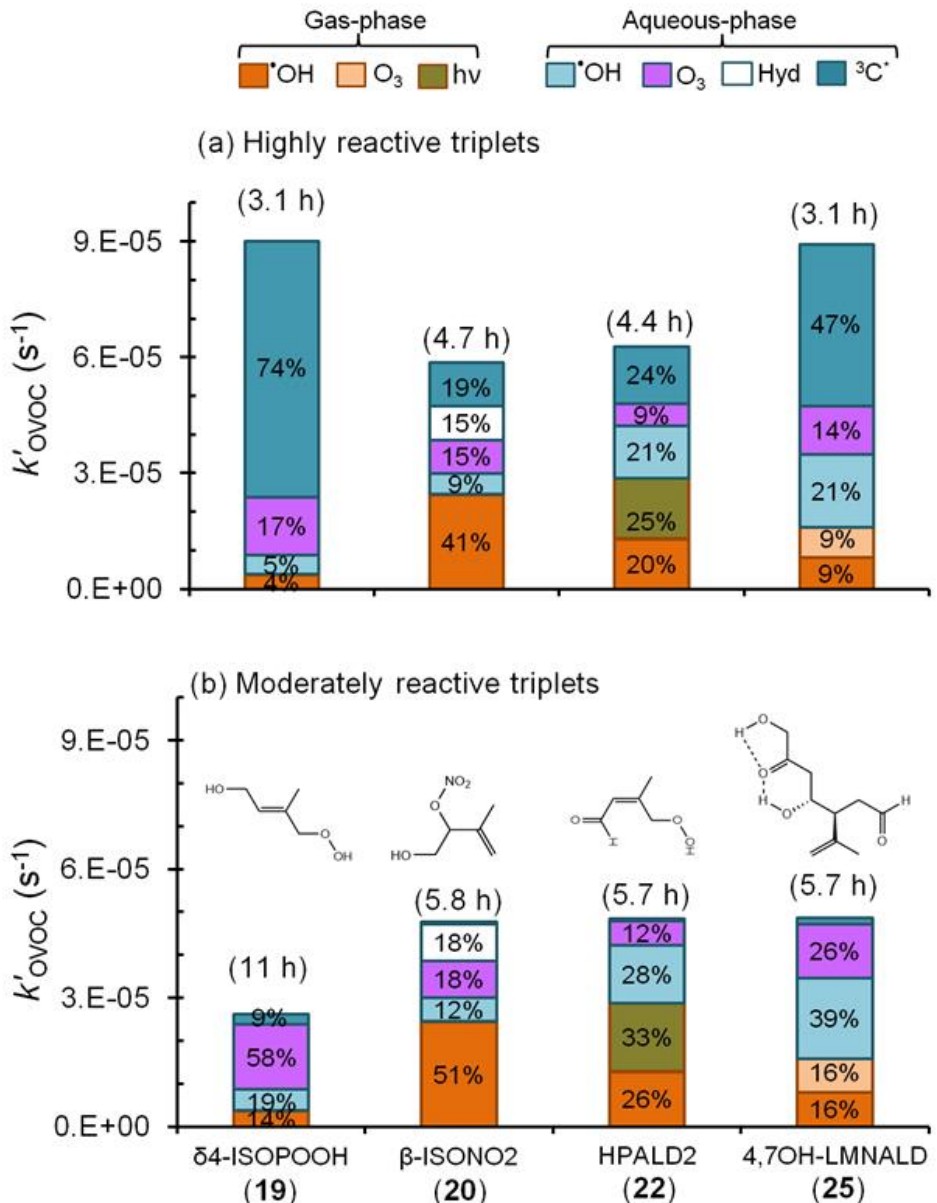

**Fig. 6** Estimated pseudo-first-order loss rate constants and corresponding lifetimes (in
parentheses) for representative isoprene- and limonene-derived oxidation products in a foggy
atmosphere (Tables S5–S7). Colors and data labels indicate the percentage of OVOC lost via
each gas and aqueous pathway, including direct photoreaction (hv) and hydrolysis (Hyd);
pathways contributing less than 4 % are not labeled. Panel (a) is a likely upper bound for the
triplet contributions to OVOC loss where we assume that all fog triplets are highly reactive, like
benzophenone. Panel (b) shows the more likely contribution from triplets, assuming moderately
reactive triplets that are more representative of the average measured in fog waters and aqueous
particle extracts(Kaur et al., 2018; Kaur and Anastasio, 2018) (Tables S5–S7).





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





**Table 1.** Measured alkene-benzophenone triplet reaction rate constants, predicted OVOC-
benzophenone triplet reaction rate constants, and computed parameters for the alkenes.

| # | Name | Abbreviation | OP [a] (V) | $\Delta G^{\ddagger}$ [b] (kcal mol$^{-1}$) | $\Delta H^{\ddagger}$ [c] (kcal mol$^{-1}$) | Measured $k_{ALK+3BP*}$ [d] ($10^8$ M$^{-1}$s$^{-1}$) |
|---|---|---|---|---|---|---|
| Alkenes | | | | | | |
| 1 | 5-Hexen-1-ol | 5HxO | 2.63 | 12.1 | 0.05 | 2.4 (0.6) |
| 2 | 2-Propen-1-ol (Allyl alcohol) | AlO | 2.65 | 10.8 | -0.04 | 2.7 (0.2) |
| 3 | 3-Hexene-1,6-diol | HDO | 2.36 | 12.3 | 0.2 | 3.1 (0.7) |
| 4 | 2,3-Butadien-1-ol | BDO | 2.46 | 10.5 | -1.5 | 3.6 (0.3) |
| 5 | 3-Buten-1-ol | 3B1O | 2.59 | 9.8 | -1.2 | 3.7 (0.5) |
| 6 | 1-Penten-3-ol | PE3O | 2.82 | 11.3 | -1.0 | 4.3 (0.4) |
| 7 | 3-Buten-2-ol | 3B2O | 2.73 | 10.6 | -1.0 | 4.9 (1.3) |
| 8 | 2-Buten-1-ol | 2B1O | 2.40 | 9.8 | -0.6 | 5.2 (1.0) |
| 9 | 5-Hexenyl acetate | 5HxAc | 2.60 | 13.7 | 2.2 | 5.9 (1.8) |
| 10 | trans-3-hexen-1-ol | tHxO | 2.28 | 12.4 | 0.03 | 14 (1) |
| 11 | 1-Chloro-3-methyl-2-butene | CMB | 2.25 | 14.1 | 2.7 | 17 (1) [e] |
| 12 | 3-Methyl-2-buten-1-ol (Prenol) | 3MBO | 2.03 | 9.7 | -1.8 | 19 (3) |
| 13 | 2-Methyl-2-penten-1-ol | 2M2PO | 2.02 | 11.6 | -1.4 | 28 (1) |
| 14 | 4-Methyl-3-penten-1-ol | 4M3PO | 1.96 | 11.5 | -0.4 | 40 (2) |
| 15 | cis-3-Hexen-1-ol | cHxO | 2.23 | 9.2 | -0.3 | 64 (6) |
| 16 | cis-3-Hexenyl acetate | cHxAc | 2.29 | 10.7 | 1.2 | 65 (6) |
| 17 | Methyl jasmonate | MeJA | - [f] | - [f] | - [f] | 75 (5) |
| Predictions for isoprene- and limonene-derived OVOCs | | | | | | Predicted $k_{OVOC+3BP*}$ [g] ($10^8$ M$^{-1}$s$^{-1}$) |
| 18 | β4-Isoprene hydroxyhydroperoxide | β4-ISOPOOH | 3.13 | 13.2 | 0.3 | 0.80 (0.18) |
| 19 | δ4-Isoprene hydroxyhydroperoxide | δ4-ISOPOOH | 2.28 | 10.5 | -2.0 | 14 (3) |
| 20 | β-Isoprene hydroxynitrate | β-ISONO2 | 2.64 | 13.2 | 1.4 | 4.1 (0.9) |
| 21 | δ-Isoprene hydroxynitrate | δ-ISONO2 | 2.40 | 10.0 | -1.9 | 9.2 (2.1) |
| 22 | Isoprene hydroperoxyaldehyde 2 | HPALD2 | 2.65 | 10.4 | -2.6 | 4.0 (0.9) |
| 23 | Limononaldehdye | LMNALD | 2.22 | 9.9 | -1.4 | 17 (4) |
| 24 | 2,5-Dihydroxy limononaldehdye | 2,5OH-LMNALD | 2.26 | 10.1 | -2.2 | 15 (3) |
| 25 | 4,7-Dihydroxy limononaldehdye | 4,7-OH-LMNALD | 2.41 | 10.6 | -0.8 | 8.9 (2.0) |

[a] One-electron oxidation potential, calculated using the CBS-QB3 compound method.
[b,c] Lowest transition state energy barrier for H-abstraction by triplet benzophenone, calculated using
uB3LYP/6-31+G(d,p).
[d] Measured bimolecular rate constant for alkene reacting with ³BP* with uncertainties of ± 1 standard
deviation, determined from triplicate measurements (Table S1 of the supplement).





[e] Listed uncertainty is ± 1 standard error, n =1.
[f] The oxidation potential and energy barriers could not be computed for MeJA (**17**). Because the CB3-
QB3 method scales at $N^7$ (where $N$ is the number of atoms), the larger compound required more
computational power than available.
[g] Predicted bimolecular rate constant for select isoprene- and limonene-derived OVOCs reacting with
$^3$BP*, determined from the correlation between OP and $k_{ALK+3BP*}$. Listed uncertainties are ± 1 standard
error propagated from the error of the slope of the quantitative structure-activity relationship between
oxidation potential and $k_{ALK+3BP*}$ (Fig. 3).