# Peer review of "Aqueous Reactions of Organic Triplet Excited States with Atmospheric"

_Atmospheric Chemistry and Physics, 2018_

## Referee Comment (RC1) · Anonymous Referee #1 · 4 Feb 2019

Major comment: The authors present a nice study on the reactions of a model triplet species with various alkenes and reveal which features (e.g. one electron reduction potential, double bond location) have a higher reactivity towards triplets. When reading the manuscript, I was curious whether or not the authors could confirm that the rate constants for triplet benzophenone are similar to those generated from brown carbon/natural organic matter (NOM). Although beyond the scope of this study, a discussion of how the rate constants for the 17 model compounds might be different for triplet NOM, or how they might vary if NOM is also present, might be useful.

Minor comments: Abstract/Intro Is brown carbon something that needs to be defined here (like in line 46)? Or is it a fairly common term in atmospheric chemistry literature? Is "a.k.a" commonly used?

[Figure]

Line 76: what are the steady-state concentrations of OH radicals and triplets? Are the concentrations of benzophenone and alkenes used in this study environmentally relevant?

Methods Why was a pH of 5.5 selected?

Does irradiating the benzophenone solution generate other oxidants? Can you confirm all reactions due to 3BP*? Similarly, do any of the test alkenes or reference compounds degrade due to direct photoreactions when BP is not present?

How does 100 uM BP and 50 uM alkene compare to brown carbon concentrations and alkene concentrations, respectively, in fog droplets/aqueous particles?

What irradiation time or times were used? Did they vary? Is oxygen consumed in sealed quartz cell during this time, impacting rates? I imagine benzophenone and NOM have different absorbance (A) spectra? It would be interesting to compare A spectra multiplied by irradiance for benzophenone and for brown carbon (or something similar to figure S1). Fig. S1 is a bit confusing showing %transmittance for the light source and not its irradiance through the filters? I think showing the irradiance the sample sees would be more useful for a comparison to solar irradiation. I imagine the photon dose the sample sees impacts the formation of triplets, can the authors confirm that this does this not matter for the competition kinetic experiments performed here?

Line 115-117: Where was the aluminum wrapped dark in relation to the irradiated sample? If the two samples are side by side there will certainly be issues since aluminum foil is a hard reflector and could increase photon dose in irradiated sample.

Results/Discussion Lines 311-333: As the authors note, adjusting 3BP* constants is uncertain, but I also now wonder if 3MAP and DMB triplets are more representative of triplets from NOM? Or is that unknown?

---

## Referee Comment (RC2) · Anonymous Referee #2 · 6 Feb 2019

In this very ambitious study, the authors measured the kinetics of oxidation of a series of alkenes by the triplet excited state of benzophenone, which they use as a model compound for triplet excited states in atmospheric waters. They then looked for correlations between the kinetic data and various properties of the alkenes, some of which were derived using density functional theory (DFT) calculations. They found a fairly good correlation between the rate constants and the one-electron oxidation potential for the alkenes, and used that to develop a quantitative structure-activity relationship (QSAR). They used the QSAR, and more DFT calculations, to infer triplet oxidation rates for several biogenically derived alkenes. Finally, they perform some estimates of the potential importance of triplet chemistry in atmospheric waters. I recommend publication in ACP after some minor points are addressed.

[Figure]

Minor comments:

- It is not mentioned in the main text how many times each kinetic experiment was repeated - I only knew this after looking at Table S1

- Can the authors discuss and provide some estimate of the error/uncertainty for the parameters derived from the DFT calculations? How does this impact the discussion of the outliers for the QSAR?

- I note from Table S1 that several different reference probes were used. The reason for this should be discussed. The reference rates and the uncertainty in those rates should be listed/discussed. Were the uncertainties included in the reported uncertainties in k, and considered in the development of the QSAR?

- Just a suggestion: Fig. 4 and some of the discussion of these calculations could be moved to the SI, since the article is already quite dense with information and this line of inquiry was ultimately inconclusive.

- A little more information about the atmospheric lifetime calculations should be provided. Are you considering repartitioning of the OVOCs between the gas and aqueous phases as the reaction proceeds? Or are the calculated rates basically initial rates?

---

## Author Comment (AC1) · 18 Mar 2019

Please note that our response is given below each question or comment from the reviewers. We have also uploaded our responses as a separate pdf file. Please note that line numbers in the revised version will be different due to changes in the manuscript.
* * *
Anonymous Referee #1

Major comment: The authors present a nice study on the reactions of a model triplet species with various alkenes and reveal which features (e.g. one electron reduction potential, double bond location) have a higher reactivity towards triplets. When reading the manuscript, I was curious whether or not the authors could confirm that the rate

constants for triplet benzophenone are similar to those generated from brown carbon/ natural organic matter (NOM). Although beyond the scope of this study, a discussion of how the rate constants for the 17 model compounds might be different for triplet NOM, or how they might vary if NOM is also present, might be useful.

Author Response: We thank this reviewer for their thorough review and detailed, helpful comments. Based on our two studies to date, NOM triplets in fog and airborne particles are about as reactive as the triplets of 3'-methoxyacetophenone (3MAP) and 3,4-dimethoxybenzaldehyde (DMB). For the few alkenes where there are rate constants for both these triplets and triplet benzophenone, the latter is approximately 25 times more reactive. This information is in Section 3.4 of the manuscript.

Minor comments: 1. Abstract/Intro Is brown carbon something that needs to be defined here (like in line 46)? Or is it a fairly common term in atmospheric chemistry literature?

Author Response: The reviewer is correct – brown carbon is a fairly common term used in atmospheric chemistry. However, taking the reviewer's question into account, we have included a brief description in the abstract in line 13.

2. Is "a.k.a" commonly used?

Author Response: This refers to line 13 (first line of the abstract). We think it is a commonly used abbreviation, but we have replaced it with "or" to avoid any confusion.

3. Line 76: what are the steady-state concentrations of OH radicals and triplets? Are the concentrations of benzophenone and alkenes used in this study environmentally relevant?

Author Response: For this study, our goal was to measure rate constants for the BP triplet with alkenes, which does not require that the triplet concentration is environmentally relevant. Since we used a relative-rate approach, initial concentrations of the reactants do not impact the outcome. But to answer the question, we estimate that 3C* concentrations in our solutions are 10–14 to 10–15 M (see answer to Q5 for more

details) which is similar to fog triplet concentrations (Kaur and Anastasio, 2018). In comparison, our alkene concentrations are probably higher, by a factor of at least 10, compared to a fog drop. (But, as stated earlier, this does not impact our determination of the rate constant.) Some hydroxyl radical (•OH) was probably generated during our experiments. However, we estimate that the •OH concentration is small and has no significant impact on our rate constants; we discuss this issue in more detail in response to question 5 below.

4. Methods. Why was a pH of 5.5 selected?

Author Response: The pH of 5.5 was based on the average pH we measured in fog waters in a recent study of 5.6 ($\pm$ 0.9) (Kaur and Anastasio, 2017). We have added this information in line 101.

5. Does irradiating the benzophenone solution generate other oxidants? Can you confirm all reactions due to 3BP*? Similarly, do any of the test alkenes or reference compounds degrade due to direct photoreactions when BP is not present?

Author Response: This is a good question. The two most likely other oxidants formed in our system are singlet oxygen (1O2*) and hydroxyl radical (•OH).

1O2* is formed by reaction of triplets with O2 (Zepp et al., 1977; Haag and Hoigné, 1986); for 3BP*, the 1O2* yield (i.e., f$\triangle$) for this reaction is 0.35 (Wilkinson et al., 1993). Based on our measured alkene decays, the triplet BP concentration in our solutions was typically 1 $\times$ 10–15 M. As described by McNeill and Canonica (2016), the singlet oxygen concentration can be estimated by

$$[1O2^*] \approx 2 \, f\triangle \, [3C^*]$$

For 3BP* this gives a singlet oxygen concentration of nearly 1 $\times$ 10–15 M. For the three alkenes (HxAc, HxO, and MeJA) where we have rate constants with both 3BP* (this work) and 1O2* (Richards-Henderson et al., 2014b), the average value of kALK+1O2* / kALK+3BP* is 4.0 $\times$ 10–4; i.e., rate constants for alkenes with triplet BP are approximately 2500 times faster than with singlet oxygen. Thus, since the concentrations of 3C* and 1O2* are likely similar in our solutions but 1O2* reacts much more slowly with alkenes, singlet oxygen should be a negligible sink for the alkenes in our experiments. We have added this idea to the end of section 2.2.

In the case of OH, we cannot estimate its formation rate or steady-state concentration, which makes it impossible to quantify its contribution to alkene loss. However, there is at least one piece of evidence that argues against OH as a significant oxidant in our samples. OH reacts with most alkenes at very similar, near diffusion-controlled, rates. For example, the second-order rate constants for OH with allyl alcohol (AlO) and methyl jasmonate (MeJA) are 6.0 × 109 M–1 s–1 (Simic et al., 1973) and 6.7 × 109 M–1 s–1 (Richards-Henderson et al., 2014a), respectively. This is a difference of only 11%. In contrast, our measured rate constant for MeJA with 3BP* is more than 30 times higher than the value for AlO with 3BP*. This suggests that OH has no significant impact on our measured rate constants.

Finally, direct photodegradation of all alkenes was examined in illuminated solutions without BP: no direct loss was detected for any of the compounds. We added this information to section 2.2.

6. How does 100 uM BP and 50 uM alkene compare to brown carbon concentrations and alkene concentrations, respectively, in fog droplets/aqueous particles?

Author Response: Dissolved organic carbon concentrations can range between 1200 and 2700 $\mu$M-C in Davis fog drops (Anastasio and McGregor, 2001; Zhang and Anastasio, 2001; Kaur and Anastasio, 2017) and can be several orders of magnitude higher in particles. As for the alkenes, we haven't seen concentrations reported, but they are probably at least 10 times lower than our concentration. However, as mentioned above, when determining rate constants with the relative rate method the species do not need to be at atmospherically relevant concentrations.

7. What irradiation time or times were used? Did they vary?

Author Response: Irradiation times were typically between 60 and 150 minutes, with the length dependent upon the reactivity of the alkene. We have included a statement about this in Section 2.2.

8. Is oxygen consumed in sealed quartz cell during this time, impacting rates?

Author Response: We do not think there was significant consumption of dissolved O2 since the solutions started saturated with air (corresponding to 284 $\mu$M of dissolved O2) and the cell was opened multiple times during illumination when aliquots were removed. If dissolved oxygen had been significantly consumed during the course of the experiments, the concentration of BP triplet would have increased since O2 is the main sink of triplets. In that case, the rate constants for loss of alkene and reference compound would have increased with illumination time. We did not observe this: the loss of alkenes and reference compounds were always first order and the slope of the ln(C/C0) vs. time plot did not change with time. Thus, our evidence indicates that oxygen was not significantly consumed during the experiments.

9. I imagine benzophenone and NOM have different absorbance (A) spectra? It would be interesting to compare A spectra multiplied by irradiance for benzophenone and for brown carbon (or something similar to figure S1).

Author Response: While these action spectra for light absorption would be interesting, whether the BP and NOM results are similar or different wouldn't have any effect on our results. This is an interesting question, but it does not fit within the scope of our study.

10. Fig. S1 is a bit confusing showing %transmittance for the light source and not its irradiance through the filters? I think showing the irradiance the sample sees would be more useful for a comparison to solar irradiation. I imagine the photon dose the sample sees impacts the formation of triplets, can the authors confirm that this does this not matter for the competition kinetic experiments performed here?

Author Response: We thank the reviewer for this feedback. We measured the irradiance of our system and used this data to revise Figure S1 (updated figure attached).

Since we are using a relative rate method, where both the reference compound and test alkene are seeing the same concentration of 3BP*, regardless of changes in lamp flux, the irradiance does not affect the second-order rate constant that we determine. While absolute loss rates of the test and reference species are affected by the photon dose, the ratio of pseudo-first-order reaction rate constants are independent of photon flux.

11. Line 115-117: Where was the aluminum wrapped dark in relation to the irradiated sample? If the two samples are side by side there will certainly be issues since aluminum foil is a hard reflector and could increase photon dose in irradiated sample.

Author Response: We appreciate the reviewer's thoughtfulness here. The aluminum-wrapped "dark" cuvette and the illuminated sample were in the same chamber but not kept side by side. The dark cuvette was placed in a corner, not in the path of the light beam so that it was subjected to the same temperature and other conditions as the illuminated sample. We have included this clarification in the Section 2.2.

12. Results/Discussion Lines 311-333: As the authors note, adjusting 3BP* constants is uncertain, but I also now wonder if 3MAP and DMB triplets are more representative of triplets from NOM? Or is that unknown?

Author Response: Based on our recent work (the only two studies that have measured triplets in atmospheric samples, as best we know), the NOM triplets in fog and PM are typically most similar to 33MAP* and 3DMB*. This is why we adjusted the 3BP* rate constants to what we would expect for an average of 33MAP* and 3DMB*. This is described in Section 3.4.

  Anonymous Referee #2

In this very ambitious study, the authors measured the kinetics of oxidation of a series

of alkenes by the triplet excited state of benzophenone, which they use as a model compound for triplet excited states in atmospheric waters. They then looked for correlations between the kinetic data and various properties of the alkenes, some of which were derived using density functional theory (DFT) calculations. They found a fairly good correlation between the rate constants and the one-electron oxidation potential for the alkenes, and used that to develop a quantitative structure-activity relationship (QSAR). They used the QSAR, and more DFT calculations, to infer triplet oxidation rates for several biogenically derived alkenes. Finally, they perform some estimates of the potential importance of triplet chemistry in atmospheric waters. I recommend publication in ACP after some minor points are addressed.

Author Response: We thank this reviewer for their thoughtful review, encouraging comments, and specific suggestions for improvement of the manuscript.

Minor comments: 1. - It is not mentioned in the main text how many times each kinetic experiment was repeated - I only knew this after looking at Table S1.

Author Response: We have added this information to Section 2.2.

2. - Can the authors discuss and provide some estimate of the error/uncertainty for the parameters derived from the DFT calculations? How does this impact the discussion of the outliers for the QSAR?

Author Response: We think the reviewer is inquiring about the CBS-QB3 method specifically, as this was the method used for calculations of BDEs, BDFEs, and OPs. In the article describing this method, the authors state that the errors for CBS-QB3 have a mean absolute deviation of 1.10 kcal/mol on a test set they used. This is comparable to other post-Hartree-Fock ab initio methods (such as MP2, a method we used to calculate HOMOs/SOMOs), which have mean absolute deviations of 0.94 -1.21 kcal/mol on the same test set. This error of roughly 1 kcal/mol corresponds to only 0.04 V in OP, so does not account for the over- or under-prediction of the outliers in Figure 3, which are off by greater amounts. We have added a description of these errors to Section

2.3.

3. I note from Table S1 that several different reference probes were used. The reason for this should be discussed. The reference rates and the uncertainty in those rates should be listed/discussed. Were the uncertainties included in the reported uncertainties in k, and considered in the development of the QSAR?

Author Response: We used several probes so that the loss rates of the test and references alkenes were similar. If the loss rates are very different it is difficult to get good rate constants for both species during the same illumination time. The reference rates and the uncertainties are given in Table S1. We have included a statement about this in Section 2.2. The standard errors in the slope and reference rate constant were propagated to obtain the uncertainties listed for each replicate in parentheses in Table S1. Since each experiment was performed in triplicate, we used the standard deviation of the mean for the QSAR Figure 3. The uncertainties were not considered in the development of the QSAR.

4. - Just a suggestion: Fig. 4 and some of the discussion of these calculations could be moved to the SI, since the article is already quite dense with information and this line of inquiry was ultimately inconclusive.

Author Response: We appreciate the reviewer's comment about the article being dense with information. However, we feel that Figure 4 provides a good example of the computational work that was performed and illustrates an interesting difference in the reactivity of the alkenes we studied. Even though the transition state structures and associated thermodynamics didn't end up being predictive of rate constants, we have kept the figure in the main text because this is an important negative result.

5. - A little more information about the atmospheric lifetime calculations should be provided. Are you considering repartitioning of the OVOCs between the gas and aqueous phases as the reaction proceeds? Or are the calculated rates basically initial rates?

Author Response: This is a good question. Since we are only providing rough estimates, we have not considered repartitioning of the OVOCs between the phases and only considered the initial rates. We added a clarifying "initial" in the first paragraph of Section 3.4.

  References for the Author's Responses:

Anastasio, C., and McGregor, K. G.: Chemistry of fog waters in California's central valley: 1. In situ photoformation of hydroxyl radical and singlet molecular oxygen, Atmos. Environ., 35, 1079-1089, 2001. Haag, W. R., and Hoigné, J.: Singlet oxygen in surface waters .3. Photochemical formation and steady-state concentrations in various types of waters, Environ. Sci. Technol., 20, 341-348, 1986. Kaur, R., and Anastasio, C.: Light absorption and the photoformation of hydroxyl radical and singlet oxygen in fog waters, Atmos. Environ., 164, 387-397, 2017. Kaur, R., and Anastasio, C.: First Measurements of Organic Triplet Excited States in Atmospheric Waters, Environ. Sci. Technol., 52, 5218-5226, 2018. McNeill, K., and Canonica, S.: Triplet state dissolved organic matter in aquatic photochemistry: Reaction mechanisms, substrate scope, and photophysical properties, Environ. Sci. Process. Impact., 18, 1381-1399, 2016. Richards-Henderson, N. K., Hansel, A. K., Valsaraj, K. T., and Anastasio, C.: Aqueous oxidation of green leaf volatiles by hydroxyl radical as a source of SOA: Kinetics and SOA yields, Atmos. Environ., 95, 105-112, 2014a. Richards-Henderson, N. K., Pham, A. T., Kirk, B. B., and Anastasio, C.: Secondary Organic Aerosol from Aqueous Reactions of Green Leaf Volatiles with Organic Triplet Excited States and Singlet Molecular Oxygen, Environmental Science & Technology, 49, 268-276, 2014b. Simic, M., Neta, P., and Hayon, E.: Reactions of hydroxyl radicals with unsaturated aliphatic alcohols in aqueous solution. Spectroscopic and electron spin resonance radiolysis study, J. Phys. Chem., 77, 2662-2667, 1973. Wilkinson, F., Helman, W. P., and Ross, A. B.: Quantum yields for the photosensitized formation of the lowest electronically excited singlet state of molecular oxygen in solution, J. Phys. Chem. Ref. Data, 22, 113-262, 1993. Zepp, R. G., Wolfe, N. L., Baughman, G. L., and Hollis, R. C.: Singlet

oxygen in natural waters, Nature, 267, 421-423, 1977. Zhang, Q., and Anastasio, C.: Chemistry of Fog Waters in California's Central Valley–ǍŤPart 3: Concentrations and Speciation of Organic and Inorganic Nitrogen, Atmos. Environ., 35, 5629-5643, 2001.

Please also note the supplement to this comment:
https://www.atmos-chem-phys-discuss.net/acp-2018-1259/acp-2018-1259-AC1-supplement.pdf

[Figure]

**Figure S1.** Comparison of the normalized irradiance from our illumination system (red line) and Davis, midday, winter solstice sunlight from the TUV model (blue line; Madronich et al. (2002)). Our illumination system irradiance was measured using a TIDAS spectrophotometer (counts cm$^{-2}$ nm$^{-1}$ s$^{-1}$) and normalized so that the area under its curve is equal to the area under the TUV actinic flux curve. Input parameters for the TUV model were: solar zenith angle: 62°, measurement altitude: 0 km, surface albedo: 0.1, aerosol optical depth: 0.235, cloud optical depth: 0.00.

**Fig. 1.**

---

## Author Response (AR1)

Response to Reviewers comments for
**Aqueous Reactions of Organic Triplet Excited States with Atmospheric Alkenes**
By Richie Kaur et. al.

Please note:
Reviewer comment is in black text.
Our response is in blue.

**Please note that line numbers in the revised version are different due to changes in the manuscript.**
* * *
**Anonymous Referee #1**

Major comment: The authors present a nice study on the reactions of a model triplet species with various alkenes and reveal which features (e.g. one electron reduction potential, double bond location) have a higher reactivity towards triplets. When reading the manuscript, I was curious whether or not the authors could confirm that the rate constants for triplet benzophenone are similar to those generated from brown carbon/ natural organic matter (NOM). Although beyond the scope of this study, a discussion of how the rate constants for the 17 model compounds might be different for triplet NOM, or how they might vary if NOM is also present, might be useful.

We thank this reviewer for their thorough review and detailed, helpful comments. Based on our two studies to date, NOM triplets in fog and airborne particles are about as reactive as the triplets of 3'-methoxyacetophenone (3MAP) and 3,4-dimethoxybenzaldehyde (DMB). For the few alkenes where there are rate constants for both these triplets and triplet benzophenone, the latter is approximately 25 times more reactive. This information is in Section 3.4 of the manuscript.

Minor comments:
1. Abstract/Intro Is brown carbon something that needs to be defined here (like in line 46)? Or is it a fairly common term in atmospheric chemistry literature?

The reviewer is correct – brown carbon is a fairly common term used in atmospheric chemistry. However, taking the reviewer's question into account, we have included a brief description in the abstract in line 13.

2. Is "a.k.a" commonly used?

This refers to line 13 (first line of the abstract). We think it is a commonly used abbreviation, but we have replaced it with "or" to avoid any confusion.

3. Line 76: what are the steady-state concentrations of OH radicals and triplets? Are the concentrations of benzophenone and alkenes used in this study environmentally relevant?

For this study, our goal was to measure rate constants for the BP triplet with alkenes, which does not require that the triplet concentration is environmentally relevant. Since we used a relative-rate approach, initial concentrations of the reactants do not impact the outcome. But to answer the question, we estimate that $^3C^*$ concentrations in our solutions are $10^{-14}$ to $10^{-15}$ M (see answer to Q5 for more details) which is similar to fog triplet concentrations (Kaur and Anastasio, 2018). In comparison, our alkene concentrations are probably higher, by a factor of at least 10, compared to a fog drop. (But, as stated earlier, this does not impact our determination of the rate constant.)

Some hydroxyl radical ($^•OH$) was probably generated during our experiments. However, we estimate that the $^•OH$ concentration is small and has no significant impact on our rate constants; we discuss this issue in more detail in response to question 5 below.

4. Methods. Why was a pH of 5.5 selected?

The pH of 5.5 was based on the average pH we measured in fog waters in a recent study of 5.6 ($\pm$ 0.9) (Kaur and Anastasio, 2017). We have added this information in line 101.

5. Does irradiating the benzophenone solution generate other oxidants? Can you confirm all reactions due to 3BP*? Similarly, do any of the test alkenes or reference compounds degrade due to direct photoreactions when BP is not present?

This is a good question. The two most likely other oxidants formed in our system are singlet oxygen ($^1O_2^*$) and hydroxyl radical ($^•OH$).

$^1O_2^*$ is formed by reaction of triplets with $O_2$ (Zepp et al., 1977; Haag and Hoigné, 1986); for $^3BP^*$, the $^1O_2^*$ yield (i.e., $f_\Delta$) for this reaction is 0.35 (Wilkinson et al., 1993). Based on our measured alkene decays, the triplet BP concentration in our solutions was typically $1 \times 10^{-15}$ M. As described by McNeill and Canonica (2016), the singlet oxygen concentration can be estimated by

$$[^1O_2^*] \approx 2 f_\Delta [^3C^*]$$

For $^3BP^*$ this gives a singlet oxygen concentration of nearly $1 \times 10^{-15}$ M. For the three alkenes (HxAc, HxO, and MeJA) where we have rate constants with both $^3BP^*$ (this work) and $^1O_2^*$ (Richards-Henderson et al., 2014b), the average value of $k_{ALK+1O2*} / k_{ALK+3BP*}$ is $4.0 \times 10^{-4}$; i.e., rate constants for alkenes with triplet BP are approximately 2500 times faster than with singlet oxygen. Thus, since the concentrations of $^3C^*$ and $^1O_2^*$ are likely similar in our solutions but $^1O_2^*$ reacts much more slowly with alkenes, singlet oxygen should be a negligible sink for the alkenes in our experiments. We have added this idea to the end of section 2.2.

In the case of $^•OH$, we cannot estimate its formation rate or steady-state concentration, which makes it impossible to quantify its contribution to alkene loss. However, there is at least one piece of evidence that argues against $^•OH$ as a significant oxidant in our samples. $^•OH$ reacts with most alkenes at very similar, near diffusion-controlled, rates. For example, the second-order rate constants for $^•OH$ with allyl alcohol (AlO) and methyl jasmonate (MeJA) are $6.0 \times 10^9$ M$^{-1}$ s$^{-1}$ (Simic et al., 1973) and $6.7 \times 10^9$ M$^{-1}$ s$^{-1}$ (Richards-Henderson et al., 2014a), respectively. This is a difference of only 11%. In contrast, our measured rate constant for MeJA with $^3BP^*$ is more than 30 times higher than the value for AlO with $^3BP^*$. This suggests that $^•OH$ has no significant impact on our measured rate constants.

Finally, direct photodegradation of all alkenes was examined in illuminated solutions without BP: no direct loss was detected for any of the compounds. We added this information to section 2.2.

6. How does 100 uM BP and 50 uM alkene compare to brown carbon concentrations and alkene concentrations, respectively, in fog droplets/aqueous particles?

Dissolved organic carbon concentrations can range between 1200 and 2700 μM-C in Davis fog drops (Anastasio and McGregor, 2001; Zhang and Anastasio, 2001; Kaur and Anastasio, 2017) and can be several orders of magnitude higher in particles. As for the alkenes, we haven't seen concentrations reported, but they are probably at least 10 times lower than our concentration. However, as mentioned above, when determining rate constants with the relative rate method the species do not need to be at atmospherically relevant concentrations.

7. What irradiation time or times were used? Did they vary?

Irradiation times were typically between 60 and 150 minutes, with the length dependent upon the reactivity of the alkene. We have included a statement about this in Section 2.2.

8. Is oxygen consumed in sealed quartz cell during this time, impacting rates?

We do not think there was significant consumption of dissolved $O_2$ since the solutions started saturated with air (corresponding to 284 µM of dissolved $O_2$) and the cell was opened multiple times during illumination when aliquots were removed. If dissolved oxygen had been significantly consumed during the course of the experiments, the concentration of BP triplet would have increased since $O_2$ is the main sink of triplets. In that case, the rate constants for loss of alkene and reference compound would have increased with illumination time. We did not observe this: the loss of alkenes and reference compounds were always first order and the slope of the $\ln(C/C_0)$ vs. time plot did not change with time. Thus, our evidence indicates that oxygen was not significantly consumed during the experiments.

9.  I imagine benzophenone and NOM have different absorbance (A) spectra? It would be interesting to compare A spectra multiplied by irradiance for benzophenone and for brown carbon (or something similar to figure S1).

    While these action spectra for light absorption would be interesting, whether the BP and NOM results are similar or different wouldn't have any effect on our results. This is an interesting question, but it does not fit within the scope of our study.

10. Fig. S1 is a bit confusing showing %transmittance for the light source and not its irradiance through the filters? I think showing the irradiance the sample sees would be more useful for a comparison to solar irradiation. I imagine the photon dose the sample sees impacts the formation of triplets, can the authors confirm that this does this not matter for the competition kinetic experiments performed here?

    We thank the reviewer for this feedback. We measured the irradiance of our system and used this data to revise Figure S1 as:

[Figure]

Since we are using a relative rate method, where both the reference compound and test alkene are seeing the same concentration of $^3BP*$, regardless of changes in lamp flux, the irradiance does not affect the second-order rate constant that we determine. While absolute loss rates of the test and reference species are affected by the photon dose, the ratio of pseudo-first-order reaction rate constants are independent of photon flux.

11. Line 115-117: Where was the aluminum wrapped dark in relation to the irradiated sample? If the two samples are side by side there will certainly be issues since aluminum foil is a hard reflector and could increase photon dose in irradiated sample.

We appreciate the reviewer's thoughtfulness here. The aluminum-wrapped "dark" cuvette and the illuminated sample were in the same chamber but not kept side by side. The dark cuvette was placed in a corner, not in the path of the light beam so that it was subjected to the same temperature and other conditions as the illuminated sample. We have included this clarification in the Section 2.2.

12. Results/Discussion Lines 311-333: As the authors note, adjusting 3BP* constants is uncertain, but I also now wonder if 3MAP and DMB triplets are more representative of triplets from NOM? Or is that unknown?

Based on our recent work (the only two studies that have measured triplets in atmospheric samples, as best we know), the NOM triplets in fog and PM are typically most similar to $^33MAP*$ and $^3DMB*$. This is why we adjusted the $^3BP*$ rate constants to what we would expect for an average of $^33MAP*$ and $^3DMB*$. This is described in Section 3.4.

**Anonymous Referee #2**

In this very ambitious study, the authors measured the kinetics of oxidation of a series
of alkenes by the triplet excited state of benzophenone, which they use as a model
compound for triplet excited states in atmospheric waters. They then looked for correlations
between the kinetic data and various properties of the alkenes, some of which
were derived using density functional theory (DFT) calculations. They found a fairly
good correlation between the rate constants and the one-electron oxidation potential
for the alkenes, and used that to develop a quantitative structure-activity relationship
(QSAR). They used the QSAR, and more DFT calculations, to infer triplet oxidation
rates for several biogenically derived alkenes. Finally, they perform some estimates
of the potential importance of triplet chemistry in atmospheric waters. I recommend
publication in ACP after some minor points are addressed.

We thank this reviewer for their thoughtful review, encouraging comments, and specific
suggestions for improvement of the manuscript.

Minor comments:
1.   - It is not mentioned in the main text how many times each kinetic experiment was
repeated - I only knew this after looking at Table S1.

We have added this information to Section 2.2.

2.   - Can the authors discuss and provide some estimate of the error/uncertainty for the
parameters derived from the DFT calculations? How does this impact the discussion
of the outliers for the QSAR?

We think the reviewer is inquiring about the CBS-QB3 method specifically, as this was
the method used for calculations of BDEs, BDFEs, and OPs. In the article describing this
method, the authors state that the errors for CBS-QB3 have a mean absolute deviation of 1.10
kcal/mol on a test set they used. This is comparable to other post-Hartree-Fock ab initio methods
(such as MP2, a method we used to calculate HOMOs/SOMOs), which have mean absolute
deviations of 0.94 -1.21 kcal/mol on the same test set. This error of roughly 1 kcal/mol
corresponds to only 0.04 V in OP, so does not account for the over- or under-prediction of the
outliers in Figure 3, which are off by greater amounts.  We have added a description of these
errors to Section 2.3.

3.   I note from Table S1 that several different reference probes were used. The reason for
this should be discussed. The reference rates and the uncertainty in those rates should
be listed/discussed. Were the uncertainties included in the reported uncertainties in k, and considered in the development of the QSAR?

We used several probes so that the loss rates of the test and references alkenes were similar. If the loss rates are very different it is difficult to get good rate constants for both species during the same illumination time. The reference rates and the uncertainties are given in Table S1. We have included a statement about this in Section 2.2.

The standard errors in the slope and reference rate constant were propagated to obtain the uncertainties listed for each replicate in parentheses in Table S1. Since each experiment was performed in triplicate, we used the standard deviation of the mean for the QSAR Figure 3. The uncertainties were not considered in the development of the QSAR.

4. - Just a suggestion: Fig. 4 and some of the discussion of these calculations could be moved to the SI, since the article is already quite dense with information and this line of inquiry was ultimately inconclusive.

We appreciate the reviewer's comment about the article being dense with information. However, we feel that Figure 4 provides a good example of the computational work that was performed and illustrates an interesting difference in the reactivity of the alkenes we studied. Even though the transition state structures and associated thermodynamics didn't end up being predictive of rate constants, we have kept the figure in the main text because this is an important negative result.

5. - A little more information about the atmospheric lifetime calculations should be provided. Are you considering repartitioning of the OVOCs between the gas and aqueous phases as the reaction proceeds? Or are the calculated rates basically initial rates?

This is a good question. Since we are only providing rough estimates, we have not considered repartitioning of the OVOCs between the phases and only considered the initial rates. We added a clarifying "initial" in the first paragraph of Section 3.4.

**References**

Anastasio, C., and McGregor, K. G.: Chemistry of fog waters in California's central valley: 1. In situ photoformation of hydroxyl radical and singlet molecular oxygen, Atmos. Environ., 35, 1079-1089, 2001.

Haag, W. R., and Hoigné, J.: Singlet oxygen in surface waters .3. Photochemical formation and steady-state concentrations in various types of waters, Environ. Sci. Technol., 20, 341-348, 1986.

Kaur, R., and Anastasio, C.: Light absorption and the photoformation of hydroxyl radical and singlet oxygen in fog waters, Atmos. Environ., 164, 387-397, 2017.

Kaur, R., and Anastasio, C.: First Measurements of Organic Triplet Excited States in Atmospheric Waters, Environ. Sci. Technol., 52, 5218-5226, 2018.

McNeill, K., and Canonica, S.: Triplet state dissolved organic matter in aquatic photochemistry: Reaction mechanisms, substrate scope, and photophysical properties, Environ. Sci. Process. Impact., 18, 1381-1399, 2016.

Richards-Henderson, N. K., Hansel, A. K., Valsaraj, K. T., and Anastasio, C.: Aqueous oxidation of green leaf volatiles by hydroxyl radical as a source of SOA: Kinetics and SOA yields, Atmos. Environ., 95, 105-112, 2014a.

Richards-Henderson, N. K., Pham, A. T., Kirk, B. B., and Anastasio, C.: Secondary Organic Aerosol from Aqueous Reactions of Green Leaf Volatiles with Organic Triplet Excited States and Singlet Molecular Oxygen, Environmental Science & Technology, 49, 268-276, 2014b.

Simic, M., Neta, P., and Hayon, E.: Reactions of hydroxyl radicals with unsaturated aliphatic alcohols in aqueous solution. Spectroscopic and electron spin resonance radiolysis study, J. Phys. Chem., 77, 2662-2667, 1973.

[revised manuscript text omitted]

This supporting information contains: 19 Pages, 8 Tables, and 9 Figures

Submitted to Atmospheric Chemistry and Physics, 2 December, 2018

**Table S1.** Reference probes and triplicate measurements of rate constants for alkenes in a solution at pH 5.5. Errors (in parentheses) for each replicate measurement represent ± 1 standard error, determined by propagating errors in the slope of the relative rate plot and in the reference compound rate constant. Errors on the average values represent ± 1 σ determined from the average of the replicate values.

| # | Alkene Name (ALK) | Abbreviation | Reference Probe | $k_{ALK+3BP*}$ ($10^8$ M$^{-1}$ s$^{-1}$) | | | |
|---|---|---|---|---|---|---|---|
| | | | | Replicate 1 | Replicate 2 | Replicate 3 | Average (SD) |
| 1 | 5-Hexen-1-ol | 5HxO | 3MBO | 3.1 (0.4) | 1.9 (0.3) | 2.2 (0.3) | 2.4 (0.6) |
| 2 | Allyl alcohol | AlO | BDO | 2.8 (0.3) | 2.5 (0.2) | 2.8 (0.3) | 2.7 (0.2) |
| 3 | 3-Hexene-1,6-diol | HDO | 3MBO | 2.5 (0.4) | 3.7 (0.5) | 3.2 (0.4) | 3.1 (0.7) |
| 4 | 2,3-Butadien-1-ol | BDO | 3MBO | 3.3 (0.5) | 3.8 (0.5) | 3.6 (0.5) | 3.6 (0.3) |
| 5 | 3-Buten-1-ol | 3B1O | cHxO | 4.2 (0.4) | 3.2 (0.3) | 3.6 (0.4) | 3.7 (0.5) |
| 6 | 1-Penten-3-ol | PE3O | 3B2O | 3.9 (1.1) | 4.2 (1.2) | 4.7 (1.3) | 4.3 (0.4) |
| 7 | 3-Buten-2-ol | 3B2O | cHxO | 5.7 (0.5) | 5.6 (0.5) | 3.3 (0.5) | 4.9 (1.3) |
| 8 | 2-Buten-1-ol | 2B1O | 4M3PO | 5.6 (0.2) | 4.1 (0.3) | 5.9 (0.5) | 5.2 (1.0) |
| 9 | 5-Hexenyl acetate | 5HxAc | 3MBO | 4.7 (0.7) | 5.0 (0.7) | 7.9 (1.1) | 5.9 (1.8) |
| 10 | trans-3-hexen-1-ol | tHxO | 3MBO | 13 (2) | 14 (2) | 14 (2) | 14 (1) |
| 11 | 1-Chloro-3-methyl-2-butene | CMB | BDO [a] | 17 (1) | - | - | 17 (1) [b] |
| 12 | 3-Methyl-2-buten-1-ol | 3MBO | cHxO | 21 (2) | 20 (2) | 16 (1) | 19 (3) |
| 13 | 2-Methyl-2-penten-1-ol | 2M2PO | 3MBO | 29 (4) | 28 (4) | 28 (4) | 28 (1) |
| 14 | 4-Methyl-3-penten-1-ol | 4M3PO | 3MBO | 42 (6) | 39 (5) | 40 (5) | 40 (2) |
| 15 | cis-3-hexen-1-ol | cHxO | PhOH | 62 (11) [c] | 70 (13) | 59 (11) | 64 (6) |
| 16 | cis-3-hexenyl acetate | cHxAc | cHxO | 66 (7) | 71 (6) | 59 (5) | 65 (6) |
| 17 | Methyl jasmonate | MeJA | cHxO | 80 (7) | 69 (6) | 75 (7) | 75 (5) |

[a] Measurement of the rate constant for CMB was done in a solution containing a minimal amount of acetonitrile to dissolve the compound.

[b] Error represents ± 1 SE, based on the SE of the relative rate slope and reference rate constant $k_{BDO+3BP*}$ given in the table.

[c] Phenol (PhOH) was used as the reference probe using the reference rate constant of 3.9 (± 0.7) $\times$ $10^9$ M$^{-1}$ s$^{-1}$, measured in this study, using 2,4,6,-trimethylphenol (TMP) as a reference compound ($k_{TMP+3BP*}$ = 5.1 (± 0.9) $\times$ $10^9$ M$^{-1}$ s$^{-1}$; Canonica et al. (2000)).

**Table S2.** Highest- and singly-occupied molecular orbitals (HOMOs, SOMOs) of representative alkenes
showing removing of an electron from the π system.[†]

| ALK Abbreviation (#) | HOMO | SOMO | HOMO+1 (eV) | SOMO+1 (eV) |
|---|---|---|---|---|
| HDO (3) | | | 0.39 | 0.45 |
| BDO (4) | | | 0.41 | 0.45 |
| PE3O (6) | | | 0.41 | 0.49 |
| 3B2O (7) | | | 0.40 | 0.49 |
| 3MBO (12) | | | 0.38 | 0.48 |
| cHxO (15) | | | 0.39 | 0.46 |

[†] HOMOs and SOMOs were computed from single point calculations at MP2/CBSB3 (Frisch et al.,
2016). HOMO+1 and SOMO+1 values in eV are shown relative to HOMO and SOMO, respectively.

**Table S3.** HOMOs and SOMOs of alkenes showing removing of an electron from the oxygen.[†]

| ALK Abbreviation (#) | HOMO | SOMO | HOMO+1 (eV) | SOMO+1 (eV) |
|---|---|---|---|---|
| **3B1O** **(5)** |  |  |  *0.40* |  *0.45* |
| **HxAc** **(16)** |  |  |  *0.39* |  *0.44* |
| **MeJA** **(17)** |  |  |  *0.37* |  *0.19* |

[†]HOMOs and SOMOs were computed from single point calculations at MP2/CBSB3 (B3LYP/CBSB7 was used for ALKs **16** and **17**). HOMO+1 and SOMO+1 values in eV are shown relative to HOMO and SOMO, respectively.

| 36 | **Table S4.** Oxidation potentials (in units of V) of various isomers of isoprene- and limonene-derived |
| 37 | OVOCs, calculated using the CBS-QB3 compound method. The lowest energy isomer for each OVOC |
| 38 | is highlighted using a blue box. Compounds not shown here (18, 20 and 22) have no relevant isomers. |

| δ4 ISOPOOH (19) | δISONO2 (21) | LMNALD (23) | 2,5OH-LMNALD (24) | | 4,7OH-LMNALD (25) |
|---|---|---|---|---|---|
| 2.25 | 2.40 | 2.22 | 2.06 | 2.24 | 2.17 |
| 2.28 | 2.45 | 2.28 | 2.12 | 2.26 | 2.17 |
| | | | 2.23 | 2.39 | 2.21 |
| | | | 2.24 | 2.39 | 2.41 |
| | | | | | 2.48 |

| 39 |

**Table S5.** Measured or estimated rate constants for reactions of OVOCs with oxidants, photolysis, and hydrolysis.

| OVOC | | Gas-phase rate constants (cm$^3$ mlc$^{-1}$ s$^{-1}$) | | | | | |
| --- | --- | --- | --- | --- | --- | --- | --- |
| # | Name | $k_{OVOC+OH}$ | Reference | $k_{OVOC+O3}$ | Reference | $j_{Photolysis}$ (s$^{-1}$) | Reference |
| 18 | β4-ISOPOOH | 1.2E-10 | St. Clair et al. (2015) | 1.3E-17 | Khamaganov and Hites (2001) | | |
| 19 | δ4-ISOPOOH | 1.2E-10 | St. Clair et al. (2015) | 1.3E-17 | Khamaganov and Hites (2001) | | |
| 20 | β-ISONO2 | 5.4E-11 | Lee et al. (2014) | 5.0E-19 | Lee et al. (2014) | | |
| 21 | δ-ISONO2 | 1.1E-10 | Lee et al. (2014) | 2.8E-17 | Lee et al. (2014) | | |
| 22 | HPALD2 | 5.1E-11 | Wolfe et al. (2012) | 1.2E-18 | Wolfe et al. (2012) | 6.3E-05 | Wolfe et al. (2012) |
| 23 | LMNALD | 1.6E-10 | Gill and Hites (2002) | 2.1E-16 | Khamaganov and Hites (2001) | | |
| 24 | 2,5OH-LMNALD | 1.6E-10 | Gill and Hites (2002) | 2.1E-16 | Khamaganov and Hites (2001) | | |
| 25 | 4,7OH-LMNALD | 1.6E-10 | Gill and Hites (2002) | 2.1E-16 | Khamaganov and Hites (2001) | | |
| 26 | HPALD1 | 5.1E-11 | Wolfe et al. (2012) | 1.2E-18 | Wolfe et al. (2012) | 6.3E-05 | Wolfe et al. (2012) |
| **OVOC** | | **Aqueous-phase rate constants (L mol$^{-1}$ s$^{-1}$)** | | | | | |
| # | Name | $k_{OVOC+OH}$ | Reference | $k_{OVOC+O3}$ | Reference | $k'_{Hydrolysis}$ (s$^{-1}$) | Reference |
| 18 | β4-ISOPOOH | 2.5E+09 | Rivera-Rios et al. (2018) | 4.7E+04 | Schöne and Herrmann (2014) [a] | | |
| 19 | δ4-ISOPOOH | 2.5E+09 | Rivera-Rios et al. (2018) | 4.7E+04 | Schöne and Herrmann (2014) [a] | | |
| 20 | β-ISONO2 | 5.0E+09 | Herrmann et al. (2015) [b] | 4.7E+04 | Schöne and Herrmann (2014) [a] | 1.6E-05 | Jacobs et al. (2014) |
| 21 | δ-ISONO2 | 5.0E+09 | Herrmann et al. (2015) [b] | 4.7E+04 | Schöne and Herrmann (2014) [a] | 6.8E-03 | Jacobs et al. (2014) |
| 22 | HPALD2 | 9.0E+09 | Schöne et al. (2014) [c] | 2.3E+04 | Schöne and Herrmann (2014) [c] | | |
| 23 | LMNALD | 1.0E+10 | Witkowski et al. (2018) [d] | 4.0E+04 | Witkowski et al. (2018) [d] | | |
| 24 | 2,5OH-LMNALD | 1.0E+10 | Witkowski et al. (2018) | 4.0E+04 | Witkowski et al. (2018) [d] | | |
| 25 | 4,7OH-LMNALD | 1.0E+10 | Witkowski et al. (2018) [d] | 4.0E+04 | Witkowski et al. (2018) [d] | | |
| 26 | HPALD1 | 9.0E+09 | Schöne et al. (2014) [c] | 2.3E+04 | Schöne and Herrmann (2014) [c] | | |

[a] Average of rate constants for methacrolein and methyl vinyl ketone used as a proxy.
[b] Estimate based on the rate constants for similar unsaturated compounds with •OH in the indicated reference.
[c] Rate constant for methacrolein used as a proxy.
[d] Rate constants for neutral dicarbonyl derivatives of  limonic and limononic acids, used a proxies.

**Table S6.** Loss rate constants for OVOCs due to different pathways.

| # | OVOC Name | $K_H^a$ (M atm$^{-1}$) | $\chi_{aq}^b$ | Pseudo-first-order rate constant for loss due to oxidants in the gas-phase (s$^{-1}$) | | | Pseudo-first-order rate constant for loss due to oxidants in the aqueous-phase (s$^{-1}$) | | | | |
|---|---|---|---|---|---|---|---|---|---|---|---|
| | | | | $k'_{OH,g}^c$ | $k'_{O3,g}^d$ | $j_{hv}^e$ | $k'_{OH,aq}^f$ | $k'_{O3,aq}^g$ | $k'_{3BP*,aq}^h$ (High Triplet Reactivity) | $k'_{3C*,aq}^i$ (Typical Triplet Reactivity) | $k'_{Hyd}^j$ |
| 18 | β4-ISOPOOH | 1.5E+06 | 0.97 | 1.2E-04 | 9.6E-06 | | 5.0E-06 | 1.6E-05 | 4.0E-06 | 1.4E-07 | |
| 19 | δ4-ISOPOOH | 1.2E+06 | 0.97 | 1.2E-04 | 9.6E-06 | | 5.0E-06 | 1.6E-05 | 6.8E-05 | 2.4E-06 | |
| 20 | β-ISONO2 | 5.1E+04 | 0.55 | 5.4E-05 | 3.7E-07 | | 1.0E-05 | 1.6E-05 | 2.1E-05 | 7.3E-07 | 1.6E-05 |
| 21 | δ-ISONO2 | 4.3E+04 | 0.51 | 1.1E-04 | 2.1E-05 | | 1.0E-05 | 1.6E-05 | 4.6E-05 | 1.6E-06 | 6.8E-03 |
| 22 | HPALD2 | 1.2E+05 | 0.75 | 5.1E-05 | 8.9E-07 | 6.3E-05 | 1.8E-05 | 7.6E-06 | 2.0E-05 | 7.0E-07 | |
| 23 | LMNALD | 4.5E+03 | 0.10 | 1.6E-04 | 1.6E-04 | | 2.0E-05 | 1.3E-05 | 8.3E-05 | 2.9E-06 | |
| 24 | 2,5OH-LMNALD | 8.0E+05 | 0.95 | 1.6E-04 | 1.6E-04 | | 2.0E-05 | 1.3E-05 | 7.3E-05 | 2.6E-06 | |
| 25 | 4,7OH-LMNALD | 8.0E+05 | 0.95 | 1.6E-04 | 1.6E-04 | | 2.0E-05 | 1.3E-05 | 4.4E-05 | 1.6E-06 | |
| 26 | HPALD1 | 1.2E+05 | 0.75 | 5.1E-05 | 8.9E-07 | 6.3E-05 | 1.8E-05 | 7.6E-06 | - [k] | - [k] | |

[a] Henry's law constants calculated using EPISuite version 4.1 (US EPA. Estimation Programs Interface Suite™ for Microsoft® Windows v 4.1, 2016).

[b] Fraction of OVOC in the aqueous phase, calculated as $\chi_{aq} = 1/(1+1/(K_H \times L \times R \times T))$, where $K_H$ is the Henry's law constant of the OVOC, $L$ is the assumed liquid water content ($1 \times 10^{-6}$ L-aq/L-g), $R$ is the universal gas constant (0.082 L atm K$^{-1}$ mol$^{-1}$), and $T = 298$ K.

[c,d,f,g,h,i] Pseudo-first-order rate constant for loss of OVOC due to oxidation by the given oxidant in the gas or aqueous phase, calculated by multiplying the bimolecular reaction rate constant (Table S6) with the corresponding steady-state concentration of the oxidant: $[^{\bullet}OH(g)] = 1 \times 10^6$ molecules cm$^{-3}$, $[O_3(g)] = 30$ ppbv $= 7.4 \times 10^{11}$ molecules cm$^{-3}$, $[^{\bullet}OH(aq)] = 2 \times 10^{-15}$ M (estimate in typical fog drops, includes gas-to-aqueous partitioning, Kaur and Anastasio (2017)).

$[O_3(aq)] = 3.3 \times 10^{-10}$ M (based on 30 ppbv $O_3(g)$ and $K_H = 1.1 \times 10^{-2}$ M atm$^{-1}$; Seinfeld and Pandis (2012), and $[^3C^*(aq)] = 5 \times 10^{-14}$ M (average concentration measured in Davis fog, Kaur and Anastasio (2017)).

[h] Pseudo-first-order rate constant for loss of OVOC due to oxidation by highly reactive triplets such as $^3BP^*$. This was calculated using the predicted second-order rate constants $k_{OVOC+3BP*}$ (Table 1, main text) and $[^3C^*(aq)]$ given in the footnote above.

[i] Pseudo-first-order rate constant for loss of OVOC due to oxidation by triplets of typical reactivity as measured in fog and particles in Davis, CA (Kaur and Anastasio, 2017; Kaur and Anastasio, 2018). To estimate these rate constants we multiplied the predicted second-order rate constants with $^3BP^*$ ($k_{OVOC+3BP*}$) by a factor of 0.04, which is the ratio of the average of the rate constants of reaction of MeJA with $^3$3MAP* and $^3$DMB* ($2.7 \times 10^8$ M$^{-1}$ s$^{-1}$, Table S8) divided by the rate constant for MeJA with $^3BP^*$ ($7.5 \times 10^9$ M$^{-1}$ s$^{-1}$, Tables S1 and S8).

[e, j] First-order rate constants for gas-phase photolysis and aqueous hydrolysis of the OVOC, respectively (also given in Table S5).

[k] The value of $k_{ALK+3BP*}$ for HPALD1 could not be determined due to challenges with calculating its oxidation potential. Because the CB3-QB3 method scales at $N^7$ (where $N$ is the number of atoms), the larger compound required more computational power than available.

**Table S7.** OVOC lifetimes and fractions lost due to various pathways.

| **High Triplet Reactivity Scenario** | | Total | | Fraction of OVOC lost due to each pathway [c] | | | | | | |
|---|---|---|---|---|---|---|---|---|---|---|
| # | OVOC Name | $k'_{OVOC}$ [a] $(s^{-1})$ | $\tau$ [b] (h) | •OH(g) | $O_3$(g) | hv(g) | •OH(aq) | $O_3$(aq) | $^3BP^*$(aq) | Hyd(aq) |
| 18 | β4-ISOPOOH | 2.7E-05 | 10 | 13% | 1.0% | 0% | 18% | 54% | 14% | 0% |
| 19 | δ4-ISOPOOH | 9.0E-05 | 3.1 | 3.9% | 0.32% | 0% | 5.4% | 17% | 74% | 0% |
| 20 | β-ISONO2 | 5.9E-05 | 4.7 | 41% | 0.28% | 0% | 9.4% | 15% | 19% | 15% |
| 21 | δ-ISONO2 | 3.6E-03 | 0.078 | 1.5% | 0.29% | 0% | 0.14% | 0.22% | 0.66% | 97% |
| 22 | HPALD2 | 6.3E-05 | 4.4 | 20% | 0.35% | 25% | 21% | 9.1% | 24% | 0% |
| 23 | LMNALD | 3.0E-04 | 0.93 | 49% | 47% | 0% | 0.67% | 0.44% | 2.8% | 0% |
| 24 | 2,5OH-LMNALD | 1.2E-04 | 2.4 | 6.9% | 6.7% | 0% | 16% | 11% | 59% | 0% |
| 25 | 4,7OH-LMNALD | 8.9E-05 | 3.1 | 9.0% | 8.8% | 0% | 21% | 14% | 47% | 0% |
| 26 | HPALD1 | 4.8E-05 | 5.8 | 27% | 0.46% | 33% | 28% | 12% | - [d] | 0% |
| **Typical Triplet Reactivity Scenario** | | Total | | Fraction of OVOC lost due to each pathway [c] | | | | | | |
| # | OVOC Name | $k'_{OVOC}$ $(s^{-1})$ | $\tau$ (h) | •OH(g) | $O_3$(g) | hv(g) | •OH(aq) | $O_3$(aq) | $^3C^*$(aq) | Hyd(aq) |
| 18 | β4-ISOPOOH | 2.4E-05 | 12 | 15% | 1.2% | 0% | 20% | 63% | 0.58% | 0% |
| 19 | δ4-ISOPOOH | 2.6E-05 | 11 | 14% | 1.1% | 0% | 19% | 58% | 9.0% | 0% |
| 20 | β-ISONO2 | 4.8E-05 | 5.8 | 51% | 0.43% | 0% | 12% | 18% | 0.84% | 18% |
| 21 | δ-ISONO2 | 3.5E-03 | 0.079 | 1.5% | 0.29% | 0% | 0.14% | 0.22% | 0.02% | 98% |
| 22 | HPALD2 | 4.8E-05 | 5.7 | 26% | 0.46% | 33% | 28% | 12% | 1.1% | 0% |
| 23 | LMNALD | 2.9E-04 | 1.0 | 50% | 49% | 0% | 0.69% | 0.45% | 0.10% | 0% |
| 24 | 2,5OH-LMNALD | 5.0E-05 | 5.6 | 16% | 16% | 0% | 38% | 25% | 4.9% | 0% |
| 25 | 4,7OH-LMNALD | 4.9E-05 | 5.7 | 16% | 16% | 0% | 39% | 26% | 3.0% | 0% |
| 26 | HPALD1 | 4.8E-05 | 5.8 | 27% | 0.46% | 33% | 28% | 12% | - [d] | 0% |

[a] Total pseudo-first order rate constant for loss of OVOC, calculated as $k'_{OVOC} = \Sigma(\chi_{aq} \times k'_{Ox,aq} + (1- \chi_{aq}) \times k'_{Ox,gas})$. All pseudo-first-order rate constants ($k'_{Ox,aq}$, $k'_{Ox,gas}$, $j_{hv}$, $k'_{Hyd}$) are given in Table S6.

[b] Total lifetime of OVOC, calculated as $1/k'_{OVOC}$.

[c] Fraction of OVOC lost due to each pathway, calculated as $(\chi_{aq} \times k'_{Ox,aq})/k'_{OVOC}$ for aqueous pathways and $((1-\chi_{aq}) \times k'_{Ox,gas})/k'_{OVOC}$ for gas-phase processes.

[d] We were unable to compute the oxidation potential for HAPLD1 (see footnote *k* in Table S6), and thus could not estimate its rate constant with triplets.

**Table S8.** Second-order rate constants for reaction of some alkenes with model triplet excited states.

| ALK | $k_{ALK+3C*}$ $10^8$ M$^{-1}$ s$^{-1}$ | | | Average $(k_{MeJA+33MAP*}, k_{MeJA+3DMB*})$ / $k_{MeJA+3BP*}$ [c] |
|---|---|---|---|---|
| | $^3$3MAP* | $^3$DMB* | $^3$BP* | |
| cHxO (**15**) | 1.1 ($\pm$ 0.2) [a] | 0.24 ($\pm$ 0.10) [a] | 64 ($\pm$ 6) [b] | 0.010 |
| cHxAc (**16**) | 7.9 ($\pm$ 2.0) [a] | 15 ($\pm$ 4) [a] | 65 ($\pm$ 6) [b] | 0.18 |
| MeJA (**17**) | 1.2 ($\pm$ 0.3) [a] | 4.1 ($\pm$ 1.6) [a] | 75 ($\pm$ 5) [b] | 0.035 [d] |

[a] Rate constants from Richards-Henderson et al. (2014). Listed uncertainties are $\pm$ 1 standard errors.
[b] Rate constants measured in this work (also shown in Table S1). Listed uncertainties here are $\pm$ 1 standard
deviation, n = 3.
[c] The ratio of the average bimolecular rate constants for reaction of MeJA with model triplets $^3$3MAP* and
$^3$DMB* to the rate constant for MeJA with $^3$BP*.
[d] This is the rate constant ratio for MeJA as well as the median value of the rate constant ratio (see footnote *a*)
for the three alkenes.

[Figure]

**Figure S1.** Comparison of the normalized irradiance from our illumination system (red line) and Davis, midday, winter solstice sunlight from the TUV model (blue line; Madronich et al. (2002)). Our illumination system irradiance was measured using a TIDAS spectrophotometer (counts cm$^{-2}$ nm$^{-1}$ s$^{-1}$) and normalized so that the area under its curve is equal to the area under the TUV actinic flux curve. Input parameters for the TUV model were: solar zenith angle: 62°, measurement altitude: 0 km, surface albedo: 0.1, aerosol optical depth: 0.235, cloud optical depth: 0.00.

[Figure]

**Figure S2.** Illustration of the relative rate technique used for measuring rate constants (Finlayson-Pitts and Pitts Jr, 1999; Richards-Henderson et al., 2014). Top panel: Aqueous loss of the alkene (4M3PO) and reference compound (3MBO) in the presence of the BP triplet under solar simulated light (298 K, pH 5.5 (± 0.2)). Bottom panel: Plot of change in concentration of reference compound against alkene. The slope represents the ratio (± 1 SE) of the bimolecular rate constants with the BP triplet.

[Figure]

**Figure S3.** Measured bimolecular rate constants of 17 alkenes with triplet benzophenone. Green bars represent biogenic volatile organic compounds known to be emitted from plants; grey bars represent other $C_3$–$C_6$ alkenes. Error bars represent $\pm 1$ standard deviation (n = 3) except for compound 11, where n = 1 and the error is $\pm$ 1 SE (see Table S1 for details). Experimental conditions: 298 K, pH 5.5 $\pm$ 0.2, 1.0 mM phosphate buffer).

[Figure]

**Figure S4.** Bond dissociation enthalpies (in italics) and bond dissociation free energies (in parentheses) in kcal mol$^{-1}$ for various hydrogens in each alkene. For each compound the hydrogen most likely to be abstracted, i.e., with the lowest bond dissociation energy, is shown in green.

[Figure]

**Figure S5.** Correlation plots for measured rate constants and various computed bond dissociation energies. (a) Log $k_{ALK+3BP*}$ versus the lowest bond dissociation enthalpy of the allylic hydrogen in each alkene (i.e., the green values in Fig. S4). (b) Log $k_{ALK+3BP*}$ versus the lowest bond dissociation free energy of the allylic hydrogen. (c) Log $k_{ALK+3BP*}$ versus the bond dissociation enthalpy of the hydrogen attached to the carbon adjacent to the –OH or –OCH$_3$ group. (d) Log $k_{ALK+3BP*}$ versus the bond dissociation free energy of the hydrogen attached to the carbon adjacent to the –OH or –OCH$_3$ group. (e): log $k_{ALK+3BP*}$ versus bond dissociation free energy of the O-H or OH$_2$C-H bond. Bond dissociation energies are shown in Fig. S4.

[Figure]

**Figure S6.** Log $k_{ALK+3BP*}$ versus lowest transition state free energy barrier. The alkenes are broken down into two groups: $k_{ALK+3BP*} < 5 \times 10^8$ M$^{-1}$ s$^{-1}$ (slow, red) and $k_{ALK+3BP*} \geq 5 \times 10^8$ M$^{-1}$ s$^{-1}$ (fast, green). The slopes ($\pm$ 1 SE) of these lines are $-$ 0.077 ($\pm$ 0.041) and $-$ 0.15 ($\pm$ 0.06) mol kcal$^{-1}$, respectively. Transition state energy barrier values are given in Table 1 of the main text. The orange line (plotted on the secondary y-axis) shows the trend in $k$ values expected from transition state theory ($k_{ALK+3BP*} = A \times \exp(-\Delta G^{\ddagger}/RT)$).

[Figure]

[Figure]

**Figure S7.** Lowest transition state energy barriers for two alkenes: **7**, 3B2O, in the top panel and **16**, cHxAc, in the bottom panel. Both show the hydrogen most likely to be abstracted during oxidation; in both cases this is an allylic H.

[Figure]

**Figure S8.** Pseudo-first-order loss rate constant of cHxO ($k^*_{cHxO}$) as a function of the
concentration of cHxO. Since these experiments were performed on different days, the values are
normalized to the photon flux of the illumination system on the day of the experiment by
dividing by $j_{2NB}$ (details in Kaur and Anastasio (2017)). The average ($\pm 1$ $\sigma$) value is $4.3 \pm 0.5$
$min^{-1}/s^{-1}$, giving a relative standard deviation of 12 %.

[Figure]

**Figure S9.** Lowest energy isomers of isoprene- and limonene-derived OVOCs, determined with gas-phase calculations using the CBS-QB3 compound method.

---

## Author Response (AR2)

Author Responses are in Blue. Please note that some line numbers may have changed due to the corrections.

**Co-Editor Decision: Publish subject to technical corrections** (18 Mar 2019) by Sergey A. Nizkorodov

Comments to the Author:

Dear authors. Congratulations on writing such an interesting paper. I suggest corrections to the final version as described below. Best regards. Sergey Nizkorodov.

We thank the Co-Editor for the approval and very helpful comments.

L57: et. al.( -> et al. (

Corrected.

Scheme 1: the subscript for the leftmost delta G should be "solv,AH", for the middle one should be "solv,A", and for the rightmost one should be "solv,H" in order to be consistent in notation with equation 3

**Corrected.**

Eq (4): is delta Gox the same as delta G rxn,aq in Scheme 1? It would be better to use the same notation in the equations and schemes.

delta Gox in Eq (4) is not the same as delta G rxn, aq in Scheme 1.

Eq (5) solve  $\rightarrow$  solv

Corrected.

L177: please provide reference for the SHE potential. I thought its value is uncertain.

We have included the reference.

L466, L543: "2" should be subscript

Corrected.

L514: is it critical to cite a paper in preparation?

This paper is now in ACPD. We have updated this citation and included the DOI. We think it is important that to cite it since it discusses our measurements of triplets in aqueous extracts of ambient particles.

L527, L638: missing page numbers

Corrected.

L558: "3" should be subscript

Corrected.

Figure S1: is it possible to provide absolute units for the for the illuminator's irradiance?

The units for the irradiance measurements are counts  $cm^{-2} s^{-1} nm^{-1}$  – we have included this in the updated figure caption. In the figure, we have normalized the irradiance of our illumination system such that the area under the curve is equal to area under the Davis winter solstice Actinic flux curve to compare the two.

[revised manuscript text omitted]